# A power law describes the magnitude of adaptation in neural populations of primary visual cortex

Elaine Tring[1], Mario Dipoppa [1] & Dario L. Ringach [1,2] ✉

How do neural populations adapt to the time-varying statistics of sensory input? We used two-photon imaging to measure the activity of neurons in mouse primary visual cortex adapted to different sensory environments, each defined by a distinct probability distribution over a stimulus set. We find that two properties of adaptation capture how the population response to a given stimulus, viewed as a vector, changes across environments. First, the ratio between the response magnitudes is a power law of the ratio between the stimulus probabilities. Second, the response direction to a stimulus is largely invariant. These rules could be used to predict how cortical populations adapt to novel, sensory environments. Finally, we show how the power law enables the cortex to preferentially signal unexpected stimuli and to adjust the metabolic cost of its sensory representation to the entropy of the environment.

Sensory systems adapt their representations to the changing statistics of the environment[1–6], integrating stimulus history over multiple timescales[7–10]. In the visual system, adaptation is distributed across multiple brain regions comprising a hierarchical network, from the retina to primary and high-level cortical areas[3,11,12]. In primary visual cortex (area V1), adaptation has been studied extensively at the single-cell level, providing a wealth of information about how tuning curves, along different dimensions, are affected by a single adapting stimulus[13–36]. Here, we adopt a complementary approach, measuring and analyzing adaptation at the level of neural populations[31] from a geometric perspective[37,38], using both simple and naturalistic stimuli[34]. We show how such a strategy allows us to derive two properties of adaptation that capture the transformation of sensory representations between environments. First, we discovered that a power law captures how the magnitudes of population responses are linked across different sensory environments. Second, we find that the directions of population responses are largely invariant. These properties clarify how adaptation generates larger responses to unexpected stimuli and maintains an efficient cortical representation[39,40].

## Results

### Describing adaptation at the population level

We developed a method to study how neural populations adapt in different environments. Consider a finite stimulus set, $S = \{s_i\}$. We define a visual environment, $A$, as one where the probability of observing $s_i$ is given by $p_A(s_i)$. To examine how neurons adapt in this environment, we present a rapid stimulus sequence by independently drawing stimuli from $p_A(s_i)$ while recording their activity. We define the vector $\mathbf{r}_A(s_i)$ as the mean response of the population over repeated presentations of $s_i$ in the environment $A$. The mean population vector is computed at the optimal time delay between stimulus and response (see Methods). Similarly, the responses of the population to the same stimulus set can be measured in a different environment, $B$, where the probability of observing a stimulus is dictated by $p_B(s_i)$. This measurement yields another mean population vector $\mathbf{r}_B(s_i)$. Given two environments, $A$ and $B$, can we describe how $\mathbf{r}_A(s_i)$ relates to $\mathbf{r}_B(s_i)$? If so, can such a model predict how the population will behave when it adapts to a new environment $C$? The main contribution of our study is to offer affirmative answers to these questions.

[1]Department of Neurobiology, David Geffen School of Medicine, University of California, Los Angeles, Los Angeles, CA 90095, USA. [2]Department of Psychology, University of California, Los Angeles, Los Angeles, CA 90095, USA. ✉e-mail: dario@ucla.edu

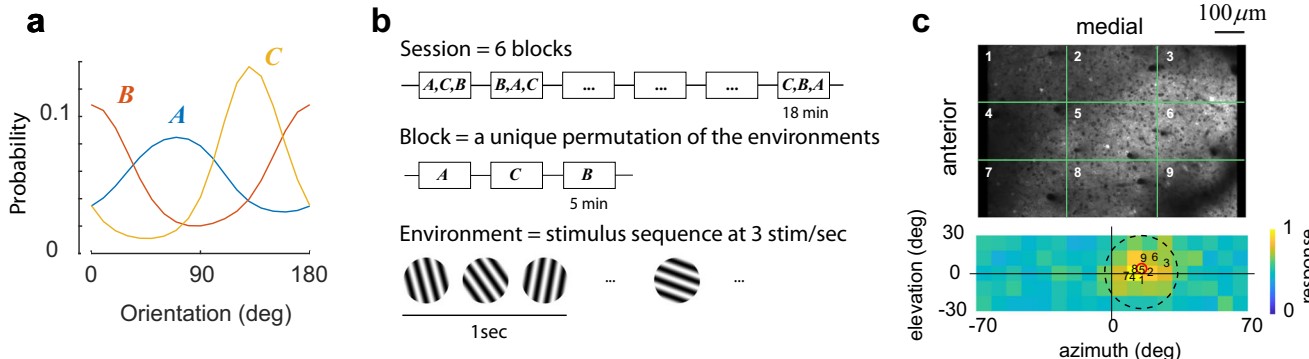

**Fig. 1 | Experimental protocol. a** Sessions included the presentation of three environments, *A*, *B*, and *C* each associated with a different distribution over a stimulus set. Stimulus sets were either sinusoidal gratings of different orientations or brief natural movie segments. **b** A session consisted of six blocks, each containing a unique permutation of all three environments. Each environment was presented for 5 min. Within an environment, stimuli were drawn from the corresponding distribution and flashed at a rate of 3/s. The presentation protocol was meant to mimic the changes of the retinal image during saccadic eye movements[18,35,51,64]. A blank screen was shown for 1 min between environments. From one session to the next, the order of the permutations was randomized. **c** Each session began with a coarse retinotopic mapping, where we determined the average locations of receptive fields within different sectors of the field of view, numbered 1–9. The bottom panel shows the center of the receptive fields for each sector mapped on the computer monitor. The background image represents the aggregate receptive field of the entire field of view. The red circle denotes its center. The dashed circle represents the circular window used during visual stimulation.

We applied this approach to study the responses of cortical populations to a set of oriented sinusoidal gratings (Fig. 1, Methods). Neural activity was recorded from layers 2/3 in primary visual cortex using in vivo, two-photon imaging. In our initial set of experiments, we used three different environments *A*, *B* and *C* defined by simple distributions. Each row of panels in Fig. 2 illustrates the outcome of one such experiment. All rows are formatted identically; thus, it suffices to describe the results from the session depicted by the panels in Fig. 2a. Here, the prior probability in environment *A* was a uniform distribution, in *B* it as a von Mises distribution with concentration $\kappa = 1.2$ centered at 0°; in *C*, the von Mises distribution was centered at the 90° (Fig. 2a(i)). The orientation tuning curve for each neuron, averaged across all environments, was computed. One can visualize the tuning curves of neurons in the population as a pseudo-color image, where each column represents a tuning curve, and cells are arranged according to their preferred orientation along the *x*-axis (Fig. 2a(ii)). Each row, therefore, represents the average population response to one orientation of the stimulus. Our analyses were based on a sub-population of well-tuned neurons (see Methods)—however, the phenomena we describe are robust to this selection (Supplementary Fig. 1).

### Response magnitudes follow a power law

We discovered that the magnitudes of responses between environments are linked via a power law. Denote by $r_X(s_i)$ the $l_2$ norm (or magnitude) of the response vector $\mathbf{r}_X(s_i)$, where *X* is one of the three possible environments $\{A,B,C\}$. Given data from two different environments, *X* and *Y*, we observed that when plotting $r_X(s_i)/r_Y(s_i)$ against $p_X(s_i)/p_Y(s_i)$, in double logarithmic coordinates, the points fall on a straight line passing through the origin (Fig. 2a(iii), solid lines indicate linear fit). In other words, the ratio between the magnitude of the responses, and the ratio between the stimulus probabilities, are related via a power law: $r_X(s_i)/r_Y(s_i) = [p_X(s_i)/p_Y(s_i)]^\beta$. The best fit for the slope was $\beta = -0.38$ (Fig. 2a(iii), inset). A goodness-of-fit measure for the linear fit we used the coefficient of determination $R^2$, which equaled 0.97, indicating that the power law provides an excellent description of the data. As the slope (also the exponent in the power law) $\beta$ is negative, the environment where a stimulus was presented more frequently generated a response with a lower magnitude—a classic signature of adaptation.

### Response directions are approximately invariant

Next, we investigated the variability in the direction of population responses to a stimulus between environments. First, we defined the unit vectors $\hat{\mathbf{r}}_X(s_i) = \mathbf{r}_X(s_i)/\|\mathbf{r}_X(s_i)\|$ and computed the resultant, $\bar{\mathbf{r}}(s_i) = \sum_{X \in \{A,B,C\}} \hat{\mathbf{r}}_X(s_i)$, representing the sum of responses across environments. Then, we calculated the cosine distance between $\hat{\mathbf{r}}_X(s_i)$ and $\bar{\mathbf{r}}(s_i)$ for all stimuli and environments. The cosine distance is defined as one minus the cosine of the angle between the vectors. The distribution of these values provides an estimate of direction scatter, which, in this experiment, had a mean $\bar{d}_s = 0.026$ (Fig. 2a(iv)). How can we tell if this value is small or large? To assess the magnitude of the scatter, we developed a "yardstick" that returns the change in the orientation of a stimulus required to cause a shift in the direction of the population response equal to scatter magnitude. We proceeded as follows. First, we computed $d(\triangle\theta)$, defined as the average cosine distance between population responses evoked by stimuli that differ by $\triangle\theta$ deg (averaged across environments). Then, we calculated the equivalent angular difference, $\triangle\theta_{eq}$, as the value of $\triangle\theta$ for which $d(\triangle\theta)$ equals the mean scatter, $\bar{d}_s$. This value is obtained from the point of intersection between the horizontal line at $\bar{d}_s$ and the $d(\triangle\theta)$ function (Fig. 2a(v)). In this case, we obtain $\triangle\theta_{eq} = 1.4$ deg. This equivalent angle is indeed small (across all experiments $\triangle\theta_{eq} = 1.26 \pm 0.41$, mean $\pm$ 1 SD), indicating that the directions of population responses are approximately invariant across environments under our experimental conditions.

### The power law predicts response magnitudes in novel environments

The measurements of responses in two environments are sufficient to obtain an estimate of the power law exponent. For example, we can use the data obtained in environments *A* and *B* to find the exponent $\beta$ that best fits the relation $r_A(s_i)/r_B(s_i) = [p_A(s_i)/p_B(s_i)]^\beta$. Suppose we now want to predict the magnitudes $r_C(s_i)$ in a new environment *C*, where the distribution of stimuli is given by $p_C(s_i)$. As the power law holds across any two environments, we must have $r_C(s_i)/r_B(s_i) = [p_C(s_i)/p_B(s_i)]^\beta$, and we can predict $r_C(s_i) = r_B(s_i)[p_C(s_i)/p_B(s_i)]^\beta$. Similarly, we can generate a prediction based on the responses in *A*, which yields $r_C(s_i) = r_A(s_i)[p_C(s_i)/p_A(s_i)]^\beta$. Finally, we can combine both estimates by taking their geometric mean:

$$r_{A,B \to C}(s_i) = \sqrt{r_A(s_i)r_B(s_i)}\left[p_C(s_i)/\sqrt{p_A(s_i)p_B(s_i)}\right]^\beta \tag{1}$$

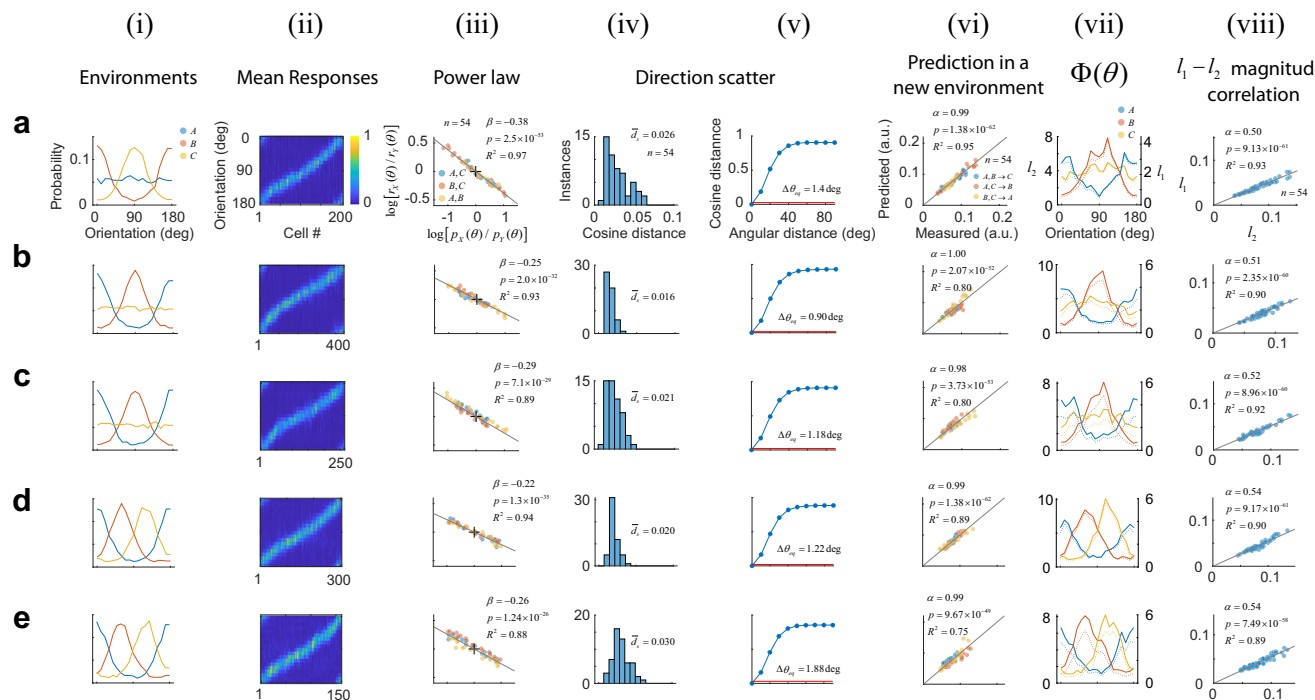

**Fig. 2 | Characterizing adaptation in neural populations.** Each row (**a**–**e**) shows results from a different experiment. Each column (**i**–**viii**) depicts separate analyses. Axes are labeled in the top row (**a**) and, unless otherwise noted, they have the same scale in all other rows (**b**–**e**). Columns represent: **i** the distribution of orientations associated with each environment. **ii** Mean responses of cells to an orientation. Each column in the image represents a tuning curve. Cells have been arranged according to their preferred orientation. Responses are normalized to the maximum and displayed according to the colormap at the inset. **iii** Logarithmic plot of the ratios between probabilities versus the ratio between magnitudes across the three possible pairs of environments. Colors indicate the corresponding pairs. The solid line represents the best-fitting line to the data (without intercept). Fit statistics appear at the inset: $\beta$ estimated slope, $p$ is the statistical significance $\beta$, $R^2$ is the goodness of fit, and $n$ is the total number of data points (the degrees of freedom of the model is $n-1$). **iv** Distribution of cosine distance scatter. The mean value appears at the inset. **v** Calculation of the equivalent angular distance. The estimated

value in each case is noted at the inset. **vi** Using the power law to predict magnitudes of population responses in a new environment. Best-fitting line (without intercept) is shown as a solid line. Fit statistics appear at the inset: $\alpha$ estimated slope, $p$ is the $p$ value of $\alpha$, $R^2$ is the goodness of fit, and $n$ is the total number of data points (the degrees of freedom of the model is $n-1$). Population magnitudes are in arbitrary units. **vii** Testing for population homeostasis. In the case of homeostasis, the function $\Phi(\theta) \cong p(\theta)r(\theta)$ should be constant (see Methods). Solid lines and left $y$-axis represent this calculation when the response magnitude is the $l_2$ norm. Dashed lines and right $y$-axis show the result if the magnitude is defined as the $l_1$ norm. **viii** Correlation between the $l_2$ and $l_1$ norms across stimuli and environments. Norms are in arbitrary units. Solid line represents best linear fit (without intercept). Fit statistics appear at the inset: $\beta$ estimated slope, $p$ is the statistical significance $\beta$, $R^2$ is the goodness of fit, and $n$ is the total number of data points (the degrees of freedom of the model is $n-1$).

The notation $r_{A,B \to C}$ indicates we are using data from environments $A$ and $B$ to predict the magnitudes of responses in $C$. As our dataset contains three environments, we can also compute $r_{A,C \to B}$ and $r_{B,C \to A}$ in a similar fashion. The measured versus predicted response magnitudes show that the power law accurately predicts the magnitudes of the responses in a novel environment, with an $R^2$ value of 0.95 (Fig. 2a(vi)). These results were remarkably consistent across experimental sessions (Fig. 2b–e). Thus, adaptation at the population level is satisfactorily captured by two simple rules: a power law for the magnitudes and the approximate invariance of the direction across environments.

### Violations of population homeostasis are common

It has been suggested that one of the goals of adaptation is to maintain population homeostasis, meaning that the system adjusts itself to keep the average firing rate of neurons constant between environments[31]. We sought to test population homeostasis in our data. It can be shown that neurons maintain a constant rate across environments if and only if the function $\Phi_X(s_i) = p_X(s_i)r_X(s_i)$ is constant for all environments $X \in \{A,B,C\}$ (see Methods). Comparing the shape of these functions between environments offers a simple test of population homeostasis. Plotting these functions immediately reveals they are far from constant (Fig. 2a(vii)). Instead, within each environment, the shape of $\Phi_X(s_i)$ resembles the shape of the probability distribution associated with $X$.

An alternative way to restate this result is to notice that homeostasis is nothing more than a power law with $\beta = -1$ (derivation in Methods). The failure of homeostasis is then evident by the fact that the experimental values of $\beta$ fall in the $[-0.4, -0.15]$ range (Supplementary Fig. 2). A similar failure of homeostasis is obtained if $r_X(s_i)$ represents the $l_1$ norm instead of the $l_2$ norm of the responses (Fig. 2a(vii), solid versus dashed lines). The reason is that, in our experiments, the $l_1$ norm is proportional to the $l_2$ norm (Fig. 2a(viii)). We will return to discuss the cause underlying this relationship below.

### Orientation distributions are represented with limited resolution

One difference between the data discussed up to this point, and an earlier study of population homeostasis[31], is that the latter used "peaked" distributions. In this condition, one orientation (the adapter) has a higher probability than the remaining orientations, which are all equally probable. Could this difference explain the lack of population homeostasis in our data? To find out, we conducted measurements in environments with peaked distributions (Fig. 3). We found that many of the results remained unchanged, including the lack of population homeostasis (Fig. 3a(vii)), the relative invariance of vector directions (Fig. 3a(iv, v)), and the strong correlation between $l_2$ and $l_1$ norms (Fig. 3(viii)). However, the power law no longer offered an adequate description of the data (Fig. 3a(iii, vi)).

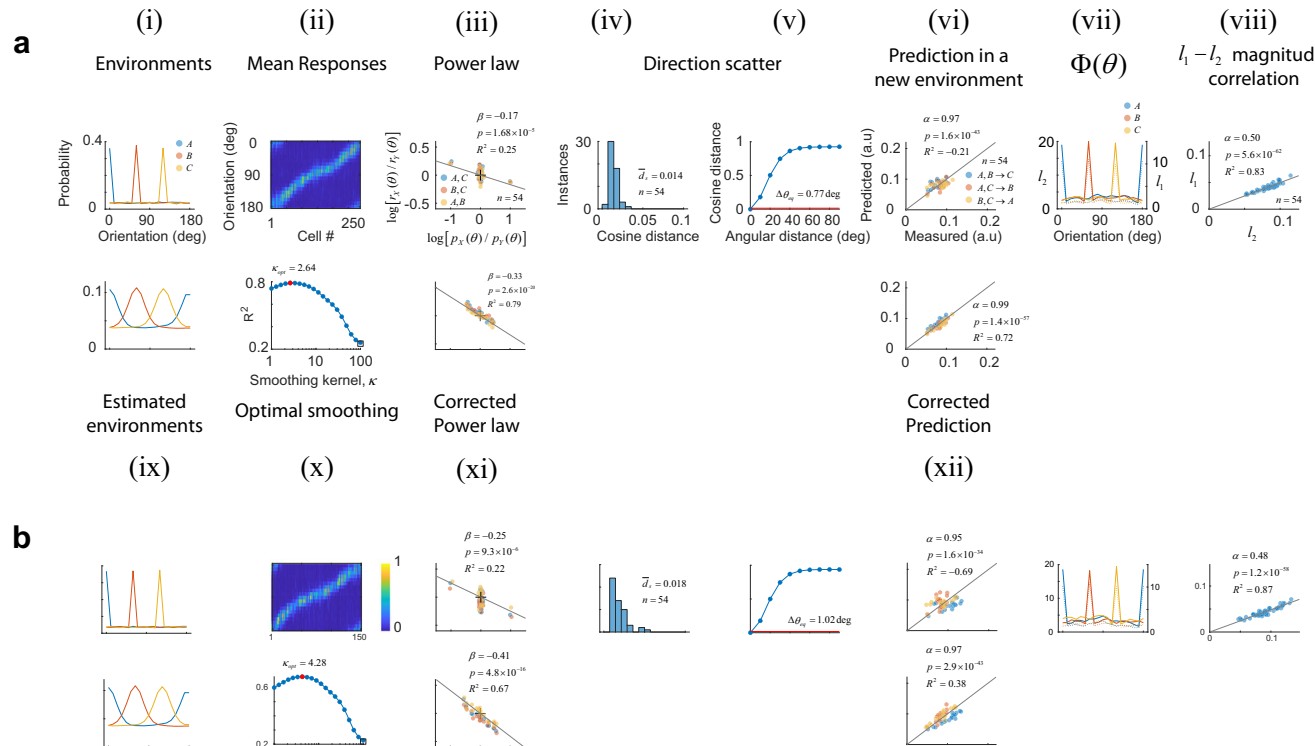

**Fig. 3 | Experimental results using peaked distributions.** Each panel (**a**, **b**) shows the result of one experiment. The top row in the panel is organized as in Fig. 2 and it shows: **i** the distribution of orientations, **ii** mean responses, **iii** ratios between probabilities versus the ratio between magnitudes across the three possible pairs of environments in double logarithmic coordinates. **iv** Distribution of cosine distance scatter, **v** equivalent angular distance, **vi** predictions of magnitudes of population responses in a new environment using the power law, **vii** test of population homeostasis, and **viii** correlation between $l_2$ and $l_2$ responses across all conditions.

The bottom rows depict: **ix** distributions obtained after smoothing the actual probabilities in (**i**) with the optimal von Mises kernel with concentration $\kappa_{opt}$, **x** goodness of fit ($R^2$) as a function of the smoothing parameter $\kappa$. The curve has an inverted U-shape with the maximum goodness of fit, shown by the red circle, attained at an intermediate value, while the open square represents the outcome without smoothing, **xi** restoration of the power law under the assumption the cortex relies on a smoothed estimate of the actual probabilities, and **xii** predictions using the power law relationship derived from (**xi**).

Seeking an explanation for this shortcoming, we reasoned that a single adaptor is not expected to influence just neurons with an orientation preference matching its orientation, but also cells with nearby orientation preferences, as their tuning bandwidths are finite. In other words, the cortex may be unable to faithfully represent a "peaked" distribution with a sharp transition between neighboring orientations. Instead, the responses may be consistent with the population behaving according to a smoothed version of the actual distribution: $p_X^\kappa(\theta) = p_X(\theta) \circledast h_\kappa(\theta)$, where $h_\kappa(\theta)$ is a von Mises kernel with concentration $\kappa$ and the operator $\circledast$ represents circular convolution. Indeed, when we repeated the analyses using $p_X^\kappa(\theta)$ instead of $p_X(\theta)$, we found there is an intermediate value of $\kappa$ that produces the best linear fit between the ratios of responses and the ratios of stimulus probabilities, as assessed by the $R^2$ measure (Fig. 3a(ix, x)). Using the optimal $\kappa$ largely restores the power law relationship (Fig. 3a(xi)). In this example, the $R^2$ for the power law linear fit is vastly improved from its original value of $-0.25$ to $+0.79$ after the smoothing correction. Similarly, the $R^2$ for the prediction of magnitudes in a new environment improved from an $R^2$ of $-0.21$ to $+0.72$ (Fig. 3a(xii)). As one may expect, smoothing did not improve the predictions when the probability distributions were smooth to begin with, like those in Fig. 2 (see Supplementary Table 1).

### Rules of adaptation apply to distributions drawn from natural images

Next, we wondered if the power law relationship would capture the data when environments are defined by a richer set of orientation distributions, such as those found in natural images (Fig. 4a). Such distributions can be skewed and have multiple peaks of varying widths

with different relative amplitudes. To answer this question, we collected data where environments were drawn from empirical distributions of natural image patches (see Methods). We were able to observe all the same phenomena (Fig. 4b–d). The goodness-of-fit could be improved to a degree by smoothing, especially in cases where one or more environments contained narrow peaks. Yet, the baseline performance of the power law and its prediction were good from the outset ($R^2$ for the power law was $0.77 \pm 0.04$, mean $\pm$ 1SD). Direction scatter remained small ($1.10 \pm 0.22$ deg, mean $\pm$1SD). A summary of the fits across all our experiments is provided in Supplementary Fig. 2 and Supplementary Table 1.

### Rules of adaptation apply to natural movie sequences

We conducted a series of experiments where the stimulus set consisted of 18 movie sequences selected from nature documentaries (Methods). Movies were randomly assigned an identification number from 1 to 18. The stimuli were not matched for luminance or contrast. We used the same environmental distributions as in our experiments with natural orientation distributions. The same phenomena could be observed under these conditions (Fig. 5a–c). The scatter in the direction of population vectors remained small, the power law still applied, and it predicted the responses in novel environments accurately. In these data, too, we observed a violation of population homeostasis. The exponents in the power law were somewhat smaller ($-0.19 \pm 0.03$, mean $\pm$1SD) compared to gratings ($-0.25 \pm 0.06$, mean $\pm$1SD) and this difference was significant (Wilcoxon rank sum test, $p = 4.4 \times 10^{-12}$). Thus, it appears that the exponent in the power law depends, in part, on the stimulus set. Notice that the two sets differed strongly in the distribution of cosine distances

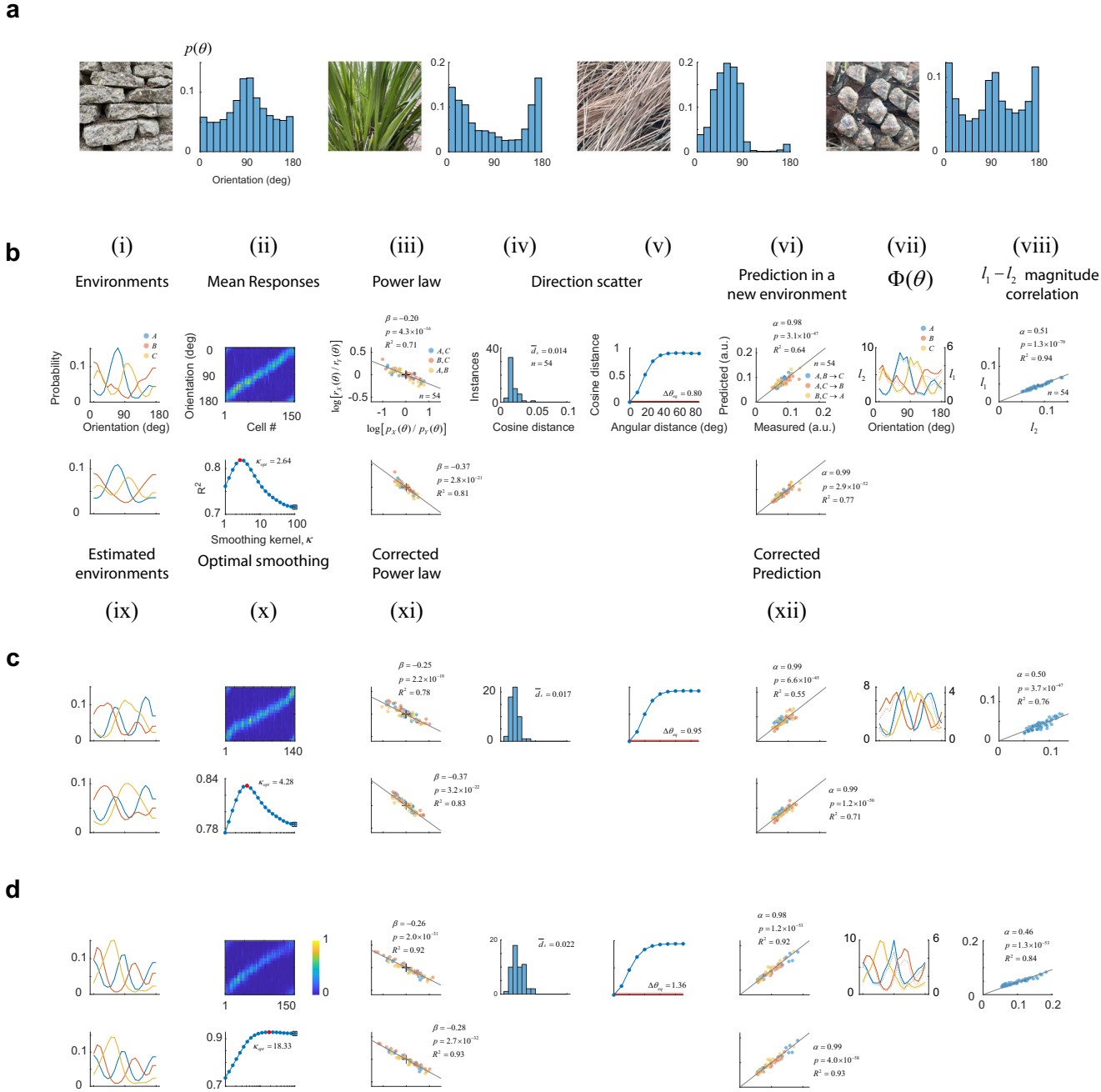

**Fig. 4 | Testing the rules of adaptation using orientation distributions drawn from natural scenes. a** Orientation distributions in natural image patches photographed by the authors on the UCLA campus. **b**–**d** Results using naturalistic environments. The panels are formatted exactly as Fig. 3, showing: **i** the distribution of orientations, **ii** mean responses, **iii** ratios between probabilities versus the ratio between magnitudes across the three possible pairs of environments in double logarithmic coordinates, **iv** Distribution of cosine distance scatter, **v** equivalent angular distance, **vi** predictions of magnitudes of population responses in a new environment using the power law, **vii** test of population homeostasis, and **viii**

correlation between $l_2$ and $l_2$ responses across all conditions. The bottom rows depict: **ix** distributions obtained after smoothing the actual probabilities in (**i**) with the optimal von Mises kernel with concentration $\kappa_{opt}$, **x** goodness of fit ($R^2$) as a function of the smoothing parameter $\kappa$. The curve has an inverted U-shape with the maximum goodness of fit attained at an intermediate value, **x** restoration of the power law under the assumption the cortex relies on a smoothed estimate of the actual probabilities, and **xii** predictions using the power law relationship derived from (**xi**).

between the responses. Movies evoked responses with pairwise cosine distances larger than 0.4, while gratings contain many pairs of stimuli differing by less than 20 deg, which evoke similar population responses with distances smaller than 0.4 (Figs. 2–4(iv)). These differences, we conjecture below, may be related to the difference in exponents obtained for different stimulus classes. Nevertheless, the power law still captured the behavior of cortical populations under adaptation to complex movie sequences.

**Adaptation is relatively fast and sensitive to spatial phase**

Finally, we investigated the dynamics of the adaptation and its dependence on spatial phase using the sinusoidal grating dataset (Fig. 6a–c). For each environment, we computed the magnitude of the responses to an orientation given that the stimulus preceding it by $T$ s differed in orientation by $\Delta\theta$, which we denote by $r(\Delta\theta,T)$. We define the modulation function as $m(\Delta\theta,T) = r(\Delta\theta,T)/r(\Delta\theta,T_\infty)$. Here, $T_\infty$ is a sufficiently large time. A plot of $m(\Delta\theta,T)$ shows that the responses have

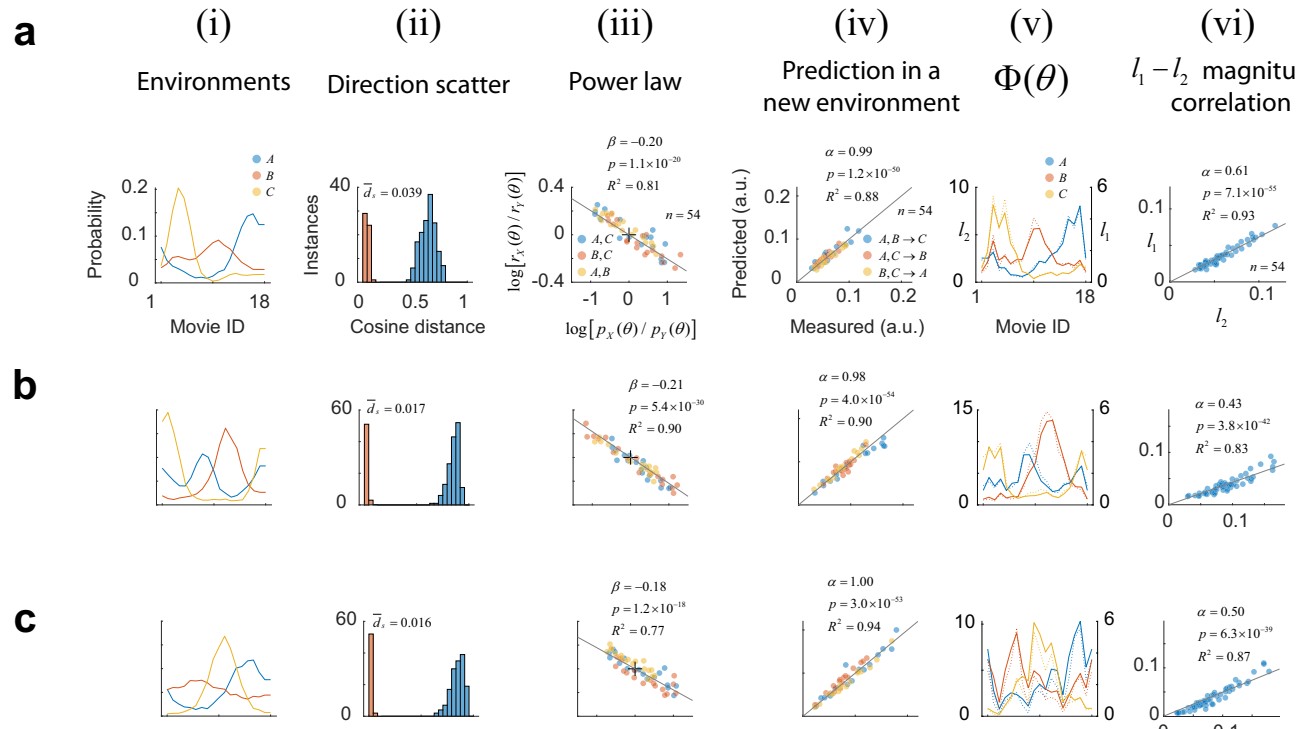

**Fig. 5 | Testing the rules of adaptation using movie sequences.** Each panel (**a**–**c**) shows the results obtained in separate experiments. Each row has the same layout. **i** Movie clips were assigned ID from 1 to 18 in a random order. Environments were defined using the same type of distributions used in the experiments described in Fig. 4ii, Direction scatter, expressed in terms of the cosine distance, is shown by the orange bars. This is the same calculation shown in Figs. 2–4(iv) for sinusoidal grating data. The blue bars show a histogram of cosine distances between the mean population responses evoked by pairs of movie clips, **iii** ratios between probabilities versus the ratio between magnitudes across the 3 possible pairs of environments in double logarithmic coordinates, **iv** scatterplot of predictions of the magnitudes of population responses in a new environment using the power law versus measured values, **v** test of population homeostasis, and **vi** correlation between the $l_2$ and $l_1$ population norms across stimuli and environments.

relatively fast dynamics. Stimuli presented more than 2sec into the past do not longer influence the magnitude of population responses[41]. We repeated a similar analysis to examine the modulatory influence of the immediately preceding stimulus as a function of both relative orientation and relative spatial phase. Clearly, the population response is sensitive to spatial phase, as the maximum suppression results when the previous stimulus matched both the orientation and phase of the present one (Fig. 6c, note that spatial phases are incongruent for different orientations).

**Computational implications of the power law**

What insights about the computational role of cortical adaptation can we gain from the power law relationship? To simplify our discussion, let us assume that in a uniform environment, $U$, where all stimuli have the same probability, the responses magnitudes are also the same. Then, according to the power law, we can write $r_X(s_i)/r_U(s_i) = [p_X(s_i)/p_U(s_i)]^{\beta}$. We are assuming $r_U(s_i) = r_U$ and $p_U(s_i) = 1/|S|$, where $|S|$ is the number of stimuli in our set. Hence, $\log r_X(s_i) = A + \beta \log p_X(s_i)$, where $A$ is a constant. From an information theoretic point of view[42], $I_X(s_i) = -\log p_X(s_i)$ represents the information content or "surprise" of observing $s_i$ in the environment $X$. We can then write $\log r_X(s_i) = A - \beta I_X(s_i)$. Thus, the logarithm of the response magnitude is linearly related to the "surprise" of observing $s_i$. Note that $\beta < 0$, so the larger the surprise the larger the response. This relationship allows us to appreciate how adaptation enables the cortex to signal unexpected, deviant, novel, or odd-ball stimuli[43–46] (all terms referring to stimuli with a low probability of appearance within an environment).

How does the metabolic cost of the representation change as a function of the predictability (or entropy) of the environment in a

population that follows a power law? We define the metabolic cost for an environment $X$ as $C_X = E\{r_X(s_i)\}$, with the expectation taken over the distribution $p_X(s_i)$. To simplify the analysis, we consider the class of environments defined by von Mises distributions. In this case, we can find close-form expressions for the entropy and metabolic cost for a population with exponent $\beta$ (Methods). In the case of a non-adapting population ($\beta = 0$) or when the population adapts perfectly ($\beta = -1$), the metabolic cost is constant with the entropy of the environment (Fig. 6d). Instead, for intermediate values $-1 < \beta < 0$, where we have partial (or "soft") adaptation, the cortex adjusts the metabolic cost of its representation to the predictability of the environment (Fig. 6e, black curve). For values of $\beta$ in the $(-0.4, -0.2)$ range, typical of our data, metabolic cost can be well approximated as a linear function of the entropy (Fig. 6d, solid curves, and Fig. 6e, red curve). Maximum modulation is attained for $\beta = -0.85$, while approximate linearity of metabolic cost with entropy is attained for $\beta > -0.5$ (Fig. 6e). In the soft adaptation regime of our data (Fig. 6e, shaded rectangle), a predictable environment (with low entropy) will be coded with a lower metabolic cost than an unpredictable one (with high entropy) and the relationship will be close to linear. These properties are consistent with the principles of efficient cortical encoding[39,40].

These findings still hold if $r_X(s_i)$ represents the $l_1$ norm rather than the $l_2$ norm in the definition of metabolic cost, as our data show the two norms are proportional to each other. What is the root of this relationship? If we assume the population contains a set of homogeneous tuning curves, the distribution of responses for any given orientation (horizontal slices through the individual panels of Fig. 2(ii)) will be the same for each orientation. Our data show that adaptation simply changes the amplitude of these vectors. Thus, for any stimulus and environment, the distribution of activity over the population will

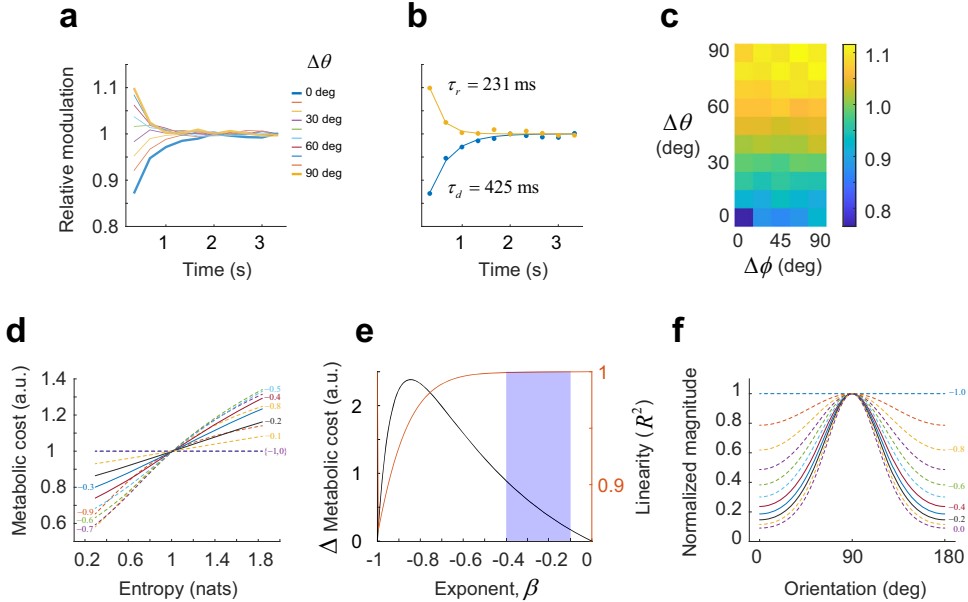

**Fig. 6 | Dynamics of adaptation and adaptation of metabolic cost to stimulus entropy. a, b** Modulation of response magnitude by a stimulus with orientation $\Delta\theta$ away, shown $T$ s earlier in the sequence. Adaptation is fast—stimuli presented beyond 2 s into the past have no influence on the population response. **b** Same data as in (**a**), for $\Delta\theta = 0$ deg and $\Delta\theta = 90$ deg with solid lines showing exponential fits to the data. We refer to $\tau_d$ as the depletion time constant and $\tau_r$ as the recovery time constant. The terms are used for convenience and are not meant to imply we know the mechanism behind adaptation is synaptic depression. **c** Modulatory effect of an immediately preceding stimulus jointly as a function of relative shifts in orientation and spatial phase. The data for $\Delta\theta = 0$ deg show that adaptation is sensitive to spatial phase. **d** Dependence of metabolic cost as a function of entropy in

environments defined by von Mises distribution with concentration parameter $\kappa \in [0,10]$. Each curve is labeled with the corresponding exponent, $\beta$. Solid curves represent the range of experimentally observed exponents. **e** Magnitude of the modulation in metabolic cost with entropy (measured by the difference between maximum and minimum values in panel (**d**)) as a function of power law exponent (solid black curve), and linearity of the cost versus entropy relationship as assessed by the $R^2$ statistic to the linear fits of the curves in panel (**d**) (red, solid curve). **f** Average population magnitudes for populations adapting with different exponents for the case of a von Mises distribution with $\kappa = 1.2$. Solid curves represent the compensation expected for the range of exponents in our experimental data. Population magnitudes are assumed to equal one in response to individual stimuli.

be one among a family of scaled distributions $f(r/s)/s$. One can easily verify this leads to norms that are proportional to each other. The constant of proportionality depends only on the shape of the distribution. A similar argument can be used for the case of natural images, assuming the tuning of the population tiles the Fourier domain and the fact that natural images have $1/f$ amplitude spectra.

Finally, to appreciate the degree to which homeostasis is achieved for the different values of $\beta$ we can plot the magnitude of the population changes as a function of orientation in the case of a von Mises distribution (Fig. 6e) (see Methods). Perfect adaptation is attained for $\beta = -1$, which generates a constant magnitude for all orientations. No adaptation corresponds to the case $\beta = 0$, where the magnitudes are modulated according to the orientation probability distribution. The range of experimental adaptation values corresponds to $\beta$ in the $(-0.4, -0.2)$ range (Fig. 6e, solid curves), which provides partial adaptation.

## Discussion

We studied the adaptation of cortical populations in different statistical environments and found that their behavior can be summarized by two properties. First, the ratio of response magnitudes to a stimulus is linked to the ratio of its prior probabilities via a power law. Second, the directions of the responses are largely invariant between environments. These relationships could be used to predict the behavior of neural populations exposed to novel environments. The same set of phenomena were obtained with environments defined by natural, orientation distributions, and with stimulus sets composed of natural image sequences. The power law seemingly failed in environments defined by peaked distributions (Fig. 3). However, the relationship could be restored using a smoothed estimate of the empirical distributions in the environment. The power law offered some insights

into the role of adaptation, revealing a neural population's ability to signal low-probability stimuli with large response magnitudes and to adjust the metabolic cost of the representation to the predictability of the environment. A limitation of the present studies is that the range of $\log\left(p_X(s_i)/p_Y(s_i)\right)$ was largely limited to $[-1, 1]$. Investigating if the power law holds for larger ranges will require increasing the experimental time or restricting the measurements to only two environments.

When our findings are compared to the seminal work of Benucci et al.[31], we find some discrepancies and some points of agreement. First, we observed a robust violation of population homeostasis in all our conditions, including environments with peaked distributions that matched their conditions (Fig. 3), where the probability of adaptor was $\sim 0.36$. Our results, instead, more closely resemble the deviations from homeostasis these authors reported when the probability of the adaptor was $0.5$ (see their Supplementary Fig. 4). It is possible that homeostasis in mice holds only when the "dynamic range" of the environments, defined as $\max_i p(s_i)/\min_i p(s_i)$, is smaller than the ones we tried so far. However, clear violations of homeostasis are also observed in environments with natural orientation distributions (Fig. 4) and with natural movie sequences (Fig. 5). Thus, we think that population homeostasis is not the sole driving force of adaptation under natural conditions. A reason for a population to adopt a "soft adaptation" regime is that, only under such conditions, the metabolic cost is adjusted as a function of the stimulus entropy. Under perfect adaptation, metabolic cost is independent of the predictability of stimuli in the environment.

Benucci et al. proposed a model where the activity is modulated by two gain factors: one applied to the tuning curves of neurons, and another applied to the population response magnitudes. They noted that the modulation of response magnitudes was the dominant

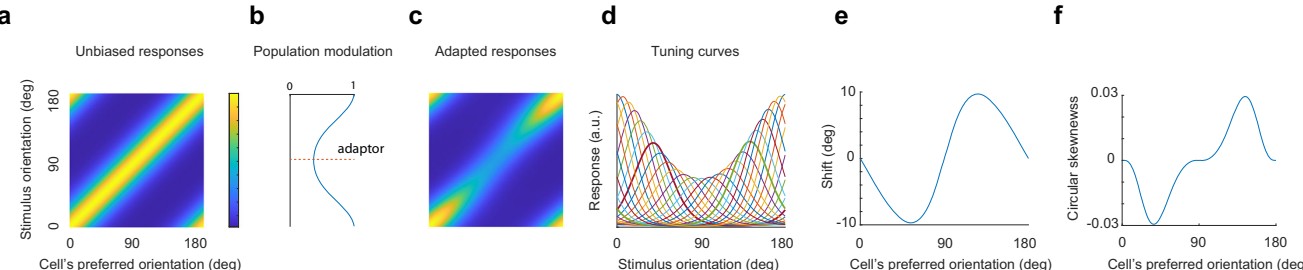

**Fig. 7 | Consistency of direction invariance with shifts in preferred orientation and tuning curve skewness. a** Responses of a homogenous population in a uniform environment. **b** Modulation function evoked by an adaptor at 90°. Each row in (**a**) is multiplied by its corresponding gain to yield the responses of the population under adaptation in (**c. d**), Examples of a few tuning curves (columns of (**c**)) under adaptation. Solid curves show two tuning curves near the adaptor. The flanks of the tuning curves closer to the adaptor fall more rapidly than those facing away, shifting their preferred orientations. **e** Shifts in the preferred orientation of tuning curves under adaptation relative to the uniform environment. **f** Circular skewness[63] of tuning curves after adaptation.

component (see their Supplementary Fig. 6). Notably, in 4 out of 5 cases where the adaptor had a probability of 0.5, a change in the gain of the population alone was sufficient to explain adaptation—there was no measurable change in the gain of the tuning curves. The dominance of the population gain is consistent with the property of direction invariance and our own supplementary analyses (Supplementary Fig. 3). Moreover, the gain functions that explained adaptation in their peaked environments were broad Gaussians, which is analogous to our finding that adaptation in the orientation domain behaves according to a smoothed distribution. The dynamics of adaptation we report (Fig. 6) are very similar to their estimates, both leading to integration windows of ~2 s (see their Supplementary Fig. 5). It is possible that our discrepancies are due to the use of different species and methodologies. They recorded multiunit activity in cat areas 17 and 18 using Utah arrays covering $4 \times 4\,mm^2$ of cortex, with electrodes arranged in a grid with 400 μm spacing. In contrast, we used single-cell, two-photon imaging in mouse primary visual cortex, with neurons located within a field of view of 730 μm × 445 μm in size. Future comparisons between imaging and electrophysiological data using the same experimental methods described here should clarify if population magnitudes are being distorted during imaging[47].

We note that the well-documented shifts in the preferred orientation of neurons away from an adapting stimulus (e.g., ref. [15]) are consistent with the modulation of population responses and direction invariance. To appreciate this point, consider a homogenous population of neural responses in a uniform environment (Fig. 7a). Here, the rows represent the population responses and the columns the tuning curves of the population. Let us assume the presence of an adaptor at 90° modulates the population responses (Fig. 7b). The cortical responses under adaptation leave the direction of population vectors invariant (Fig. 7c). If we plot the preferred orientation of the tuning curves (the columns of Fig. 7c), we notice that neurons with preferred orientations near the adaptor will have their flanks closer to the adaptor decay more rapidly than those facing away (Fig. 7d, f), causing a shift in their preferred orientation relative to the uniform environment (Fig. 7e). Thus, shifts in the preferred orientation of individual neurons between environments and the invariance of population directions coexist in our model of adaptation.

The approximate invariance of response directions allows a downstream decoder to perform well despite being "unaware"[48] of the state of adaptation of primary visual cortex. A decoder could learn a single, average map between stimulus orientation and a response direction, $\theta \rightarrow \hat{\mathbf{r}}(\theta)$. Given the direction of a new response $\hat{\mathbf{r}}_n$ in an unknown environment, the decoder could estimate the stimulus orientation as $\hat{\theta}_n = \arg\min_\theta d(\hat{\mathbf{r}}_n, \hat{\mathbf{r}}(\theta))$, where $d(\cdot,\cdot)$ is the cosine distance between the arguments. Approximate direction invariance means such a decoder is likely to perform well across different environments. We emphasize this does not mean the decoder will be

unbiased. While the scatter in response direction across environments is small, the deviations are not random and have a structure associated with the distribution in each environment that contributes to biases and changes in discriminability in unaware decoders[49]. Direction invariance under our experimental conditions is likely the result of short presentation times aimed at mimicking the changes in the retinal image due to saccadic eye movements. One may expect stronger departures from direction invariance in conditions involving longer adaptation times[50].

We observed that the exponent in the power law is stimulus dependent—on average, data obtained with movie clips were described by exponents of smaller absolute magnitude compared to gratings of different orientations. We speculate this may be due to the degree to which different stimuli generate activity in the same set of neurons. Pairs of sinusoidal gratings differing by 10° in orientation probably stimulate similar sets of neurons. This can be inferred by the fact that the average cosine distance between the population activity in this case is approximately 0.15 (Fig. 2 (v)). In contrast, the cosine distance between any two pairs of movie clips was larger than 0.5, meaning that different movies generated very different patterns and, therefore, would adapt independent set of neurons. Future experiments could explore if there is a relationship between the exponent and the pairwise cosine distance a stimulus set evokes in single populations.

The circuit implementing the power law is still under investigation. The power law behavior and its relatively fast dynamics (consistent with what has been reported in previous studies[11,35,51]) suggest a possible involvement of synaptic depression[11,16,17,35], although intrinsic cell properties could also play a role[52,53]. It is worth exploring, for example, to what extent the depression of thalamocortical synapses[54] could already generate a power law behavior. A normalization network may also be able to explain our findings[55,56], although the fact that the phenomenon is phase sensitive (Fig. 6c) does not align well with the notion that the normalization signal is pooled over many cortical neurons[57], as this would render it phase invariant. Lastly, it has not been established if adaptation can be explained exclusively as a feed-forward circuit or requires the use of top-down signals[58].

There are other important questions raised by our study. Can the model be extended to capture how the covariance of the responses and discriminability of stimuli are affected by adaptation? How does contrast sensitivity change between different environments? How do contrast and stimulus probability interact to yield a response magnitude? Can the power law describe adaptation in neural populations in other brain regions and sensory modalities? While much remains to be explored, the present analyses show that studying the responses of cortical populations at the population level can yield important insights into the signal-processing goals of adaptation and, potentially, other visual computations.

## Methods

### Experimental model and subject details

The procedures in the experiments described here were approved by UCLA's Office of Animal Research Oversight (the Institutional Animal Care and Use Committee) and were in accord with guidelines set by the U.S. National Institutes of Health. A total of four mice, one male (one) and three female, aged P35–56, were used. Animals were a cross between TRE-GCaMP6s line G6s2 (Jackson Lab, https://www.jax.org/strain/024742) and CaMKII-tTA (https://www.jax.org/strain/007004). In our analyses, we report data pooled over sex, as we did not observe any obvious differences in the exponent of the power law between male and female datasets (Supplementary Table 1, Wilcoxon test, $p = 0.18$).

### Surgery

Imaging is performed by visualizing activity through chronically implanted cranial windows over primary visual cortex. Carprofen is administered pre-operatively (5 mg/kg, 0.2 mL after 1:100 dilution). Mice are anesthetized with isoflurane (4–5% induction; 1.5–2% surgery). Core body temperature is maintained at 37.5 °C. Eyes were coated with a thin layer of ophthalmic ointment during the surgery. Anesthetized mice are mounted in a stereotaxic apparatus using blunt ear bars placed in the external auditory meatus. A portion of the scalp overlying the two hemispheres of the cortex is then removed to expose the skull. The skull is dried and covered by a thin layer of Vetbond. After the Vetbond dries (15 min), we affix an aluminum bracket with dental acrylic. The margins are sealed with Vetbond and dental acrylic to prevent any infections. A craniotomy is performed over monocular V1 on the left hemisphere using a high-speed dental drill. Special care is taken to ensure that the dura is not damaged during the process. Once the skull is removed, a sterile 3 mm diameter cover glass is placed directly on the exposed dura and sealed to the surrounding skull with Vetbond. The remainder of the exposed skull and the margins of the cover glass are sealed with dental acrylic. Mice are allowed to recover on a heating pad. When alert, they are transferred back to their home cage. Carprofen is administered post-operatively for 72 h. Mice are allowed to recover for at least 6 days before the first imaging session.

### Two-photon imaging

We conducted imaging sessions in awake animals starting 6–8 days after surgery. Mice are positioned on a running wheel and head-restrained under a resonant, two-photon microscope (Neurolabware, Los Angeles, CA) controlled by Scanbox acquisition software and electronics (Scanbox, Los Angeles, CA). The light source was a Coherent Chameleon Ultra II laser (Coherent Inc, Santa Clara, CA). Excitation wavelength was set to 920 nm. The objective was an ×16 water immersion lens (Nikon, 0.8NA, 3 mm working distance). The microscope frame rate was 15.6 Hz (512 lines with a resonant mirror at 8 kHz). The field of view was 730 μm × 445 μm. The objective was tilted to be approximately normal on the cortical surface. Images were processed using a standard pipeline consisting of image stabilization, cell segmentation and signal extraction using Suite2p (https://suite2p.readthedocs.io/)[59]. A custom deconvolution algorithm was used[60]. A summary of the experiments, including summaries of the analyses, are presented in Supplementary Table 1.

### Visual stimulation

We used a Samsung CHG90 monitor positioned 30 cm in front of the animal for visual stimulation. The screen was calibrated using a Spectrascan PR-655 spectro-radiometer (Jadak, Syracuse, NY), generating gamma corrections for the red, green, and blue components via a GeForce RTX 2080 Ti graphics card. Visual stimuli were generated by a custom-written Processing 4 sketch using OpenGL shaders (see http://processing.org). At the beginning of each experiment, we obtained a coarse retinotopy map as follows. The field of view was split into a $3 \times 3$ grid (Fig. 1c, top) and the average fluorescence in each sector was computed in real time. The screen was divided into a $5 \times 18$ grid. We randomly selected a location on the grid and presented a contrast-reversing $4 \times 4$ checkerboard for 1 sec. Within a block, each grid location was stimulated once. A total of 5 blocks were used. The data were analyzed to produce an aggregate receptive field map for each sector. The centers of each of these receptive fields are shown in the bottom panel of Fig. 1c for one experiment. The grand average of the receptive fields appears as the background in the same figure. The center of the population receptive field is used to center the location of our stimuli in these experiments. We endeavored to center the population at an elevation of around 0°, which allowed us to maximize the circular window through which we presented the stimulus (dashed circle in Fig. 1c).

The grating experiment consisted of the presentation of 100% sinusoidal gratings of 0.04 cpd using the protocol depicted in Fig. 1. The orientation domain was discretized in steps of 10°, leading to 18 orientations in the stimulus set. The spatial phases at each orientation were uniformly randomized from 0 to 360° in steps of 45°. When computing the mean population vector for a given stimulus, we averaged across spatial phases, thus minimizing the effect of eye movements on our analyses. A total of 5400 trials (6 blocks of 5 min each at 3 stim per s) were collected for each environment. For a uniform environment, this results in an average of 300 trials per orientation. For a non-uniform environment, where some orientations appeared rarely, the total number of trials could be around 50. The appearance of a new stimulus on the screen was signaled by a TTL line sampled by the microscope. As a failsafe, we also signaled the onset of the stimulus by flickering a small square at the corner of the screen. The signal of a photodiode was sampled by the microscope as well.

### Natural orientation distributions

We collected natural images from the UCLA campus (see samples in Fig. 4a). Images were converted to grayscale and the gradient of the image $\nabla I(x,y)$ was computed at each location. We computed the distribution of the magnitude of the gradient across the entire image and set a threshold at the 90th percentile of the distribution. The angle of the gradient for the pixels with magnitudes exceeding the threshold was collected. A smooth distribution in the orientation domain was then obtained via kernel density estimation[61], where the kernel was a von Mises distribution with $\kappa = 5$. The result was discretized to yield a probability distribution over angles ranging from 0 to 170 in steps of 10. A library of 150 distributions was generated by this procedure, each distribution derived from one of the natural images. In each experiment, we randomly selected three distributions and accepted them if (a) the minimum probability of a stimulus across all environments exceeded 0.03 (to ensure we would have a reasonable number of trials for all orientations) and (b) the cosine distance between the distributions was larger than 0.25 for any of the three pairs (to ensure that the distributions were sufficiently different from each other). If the test failed, we repeated the procedure until a random pick satisfied the criteria.

### Movie sequences

Movies clips were selected from high resolution nature documentaries found on the internet. Movie clips were selected to avoid transition between scenes. A stimulus set was constructed by randomly selecting 18 movie clips 333 msec in length. Thus, the size of the stimulus set was the same as that used in the grating experiments. Movie clips were ordered randomly and assigned a unique ID from 1 to 18. These movie clips were considered analogous to the set of gratings having orientations spanning 0 to 170°. To define an environment over the movies, we used the same class of probability distributions for this set as we did for natural orientation distributions. This kept some statistical features

constant across two incongruent sets of stimuli (gratings and movies), such as the entropy of the environments. The experimental session for the movies was identical to that using gratings in all respects (Fig. 1) except that instead of flashed sinusoidal gratings, the individual stimuli represented shot movie sequences.

## Optimal stimulus-response delay

For each environment, we calculated the magnitude of the population response $T$ microscope frames after the onset of the stimulus, where $T$ ranged from −2 to 15. The frame rate of the microscope was 15.53 frames/s. The time to peak of these curves agreed for all environments. In other words, the dynamics were not dependent on the statistics of the environment. We therefore averaged the magnitudes across all the three environments and defined the optimal stimulus-response delay as the time (in microscope frames) between the onset of the stimulus and the peak response magnitude of the population. This calculation was the same for gratings and for movie sequences.

## Analysis of population homeostasis

Note that the average firing rate of the population in an environment $X$ is given by $\bar{\mathbf{r}}_X = \sum_i p_X(s_i)\mathbf{r}_X(s_i)$, which can be written as $\bar{\mathbf{r}}_X = \sum_i p_X(s_i)r_X(s_i)\,\hat{\mathbf{r}}(s_i)$. Here, we have dropped the subscript from the unit vector, as directions are invariant across environments. Population homeostasis holds if the vector $\bar{\mathbf{r}}_X = \mathbf{k}$ for all environments $X$, where $\mathbf{k}$ is a constant. Assuming the vectors $\hat{\mathbf{r}}(s_i)$ are linearly independent, there is a unique way to write $\mathbf{k}$ as a linear combination of $\hat{\mathbf{r}}(s_i)$, which we write as $\mathbf{k} = \sum_i k_i\hat{\mathbf{r}}(s_i)$ (we can safely assume $\mathbf{k}$ is in the span of $\{\hat{\mathbf{r}}(s_i)\}$, otherwise, the mean rate cannot equal $\mathbf{k}$). Hence, $\bar{\mathbf{r}}_X = \sum_i p_X(s_i)r_X(s_i)\hat{\mathbf{r}}(s_i) = \sum_i k_i\hat{\mathbf{r}}(s_i)$. By coefficient matching, we conclude that homeostasis holds if and only if the function $\Phi_X(s_i) \cong p_X(s_i)r_X(s_i) = k_i$ for all environments $X$. In other words, when homeostasis holds, we have that $p_X(s_i)r_X(s_i) = k_i = p_Y(s_i)r_Y(s_i)$, which implies $r_X(s_i)/r_Y(s_i) = [p_X(s_i)/p_Y(s_i)]^{-1}$. This means that homeostasis is a particular case of the power law with $\beta = -1$.

## Entropy and metabolic cost

Consider the case where orientation is distributed according to a von Mises distribution, $p(\theta) = (1/2\pi I_0(\kappa))\exp(\kappa\cos\theta)$. The entropy of a von Mises distribution is known to be $H(\kappa) = \ln(2\pi I_0(\kappa)) - \kappa I_1(\kappa)/I_0(\kappa)$. The cost can be calculated as $C(\kappa) = E\{r(\theta)\} = E\{A\,p(\theta)^\beta\} = 2\pi A I_0((1+\beta)\kappa)/(2\pi I_0(\kappa))^{1+\beta}$. Here, $I_\nu(\bullet)$ represents the modified Bessel function of the first kind. We selected the gain $A$ so that for the cost was equal to one for an environment with an entropy of one. To find out how cost depends on entropy we plot the parametric curve $(H(\kappa),C(\kappa))$ for $\kappa \in [0,10]$ (Fig. 6d).

Incidentally, we point out that there is one measure of activity that is a linear function of the entropy of the environment. We previously showed that $\log r_X(s_i) = A - \beta I_X(s_i)$, where $I_X(s_i)$ is the surprise of stimulus $s_i$. If we take expectations on both sides, we obtain $E\{\log r_X(s_i)\} = A - \beta E\{\log p_X(s_i)\} = A - \beta H_X$, where $H_X$ is the entropy of the environment $X$.

## Statistics and reproducibility

We conducted experiments by independently measuring the adaptation of neural populations in the visual cortex in 23 different instances (see Supplementary Table 1). The goodness of fit of linear models was evaluated using the $R^2$ statistic (the coefficient of determination[62]). Results were statistically significant ($p$ values less than $10^{-4}$) and consistent across individual experiments. As the study did not involve different groups undergoing different treatments, there was no need for randomization or blind assessment of outcomes. Data selection was used to analyze neurons that showed well-tuned behavior with a circular variance[63] of less than 0.5. However, this choice has little effect on the outcome of our analyses. The same results are obtained if we, instead, work with the entire population (see Supplementary Fig. 1).

## Reporting summary

Further information on research design is available in the Nature Portfolio Reporting Summary linked to this article.

## Data availability

Data including the mean responses of the population for each experiment can be found in the Figshare repository at https://figshare.com/s/47275dd78f37230f7c83.

## Code availability

Sample code describing the structure of the database and the replication of some of our analyses can be found along with the data at https://figshare.com/s/47275dd78f37230f7c83.

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

## Acknowledgements

This work was supported by NIH grant RO1-NS116471 (D.L.R.) and RO1-EY034488 (D.L.R.) We thank Dean Buonomano, Andrea Benucci and Matteo Carandini for comments on an earlier version of this manuscript.

## Author contributions

E.T. performed all the animal surgeries. M.D. contributed to manuscript preparation, experimental design, and the conceptual and theoretical implications of the data. D.L.R. devised the experiments, wrote the visual stimulus, collected, and analyzed the data, prepared the data for the repository, and wrote the initial version of the manuscript.

## Competing interests

The authors declare no competing interests.
