## [Peer Review File · Nature Communications]

A power law describes the magnitude of adaptation in neural populations of primary visual cortexREVIEWER COMMENTS

Reviewer #1 (Remarks to the Author):

This paper presents a beautiful theoretical analysis of adaptation in neural population responses. Specifically, it shows that --- across a wide variety of stimulus environments --- the ratio between the amplitudes of the population responses to a given stimulus between two environments is a power law of the ratio between the probabilities of that stimulus in the two environments. A power of Beta = -1 would correspond to "perfect adaptation", where the total number of spikes (or "RMS spikes", depending on the definition of amplitude) elicited by each stimulus is constant across environments, while a power of Beta = 0 would correspond to "no adaptation", where the response amplitudes do not change across environments. Here the authors report a power of Beta in the range [-0.4, -0.15], where specific value depends somewhat on the details of the stimulus set. Thus, real neural populations sit somewhere short of perfect adaptation, with the consequence (as the paper indeed shows) adaptation does NOT achieve population level homeostasis. Additionally, the authors report that the angle of the population response vector does *not* change meaningfully across adaptive states.

This is a stunning set of results, representing an experimental and theoretical tour de force. The paper is clearly written and the figures are beautiful (with the exception of the small font size!). It is one of the best papers I've read all year, and I'm very excited to see it in print. I have only one major comment.

Major comment:

=====

My only major criticism is that I am not convinced that the information theoretic analysis in Fig 6 (panel d) makes sense, or adds much of value to the paper.

It seems fine to point out that the power law relationship implies that the log amplitude increases as a function of "surprise". However, the logic and motivation of this section is otherwise a bit confusing.

Recall that we are focusing on only 1 dimension of the high dimensional population activity: the L1 or L2 norm of the population response vector. Presumably the stimulus identity itself is encoded very accurately by the angle of the response vector (which as the paper shows, does not vary with the environment). Thus we are NOT talking about the encoding of the stimulus itself. But then what kind of

information are we talking about? Is it information about $P_x(S)$? (But isn't that something we could get more accurate information about by keeping track of the response vector angles?) Is it obvious that this is something the brain cares about or needs to encode? If the authors do indeed mean information about the adaptive state or environment, then it would be more useful to ask questions about how you might decode this information, rather than merely quantifying it with bits or KL divergence.

Fig 6D focuses on the KL divergence between two stimulus distributions, and showing that we can extract this from the expected log ratio of response amplitudes. However, this seems like a mathematical curiosity that follows straightforwardly from power law adaptation and the definition of KL divergence, and I'm not convinced that this is relevant or meaningful. Can the authors offer a more convincing argument for why this would matter? (Do they wish to argue that the brain needs to compute the KL divergence between different distributions over stimuli for some reason? Or is this quantity useful in some way for a computation the brain needs to perform?)

In its current form, this analysis feels like a bit of an afterthought, and as a result it seems to me to detract from rather than add to the main message. So I would suggest that the authors either motivate and explain the information-theoretic argument more fully and clearly or else feel free to remove it. (I think the paper stands quite nicely on its own even without it!)

=====

Detailed comments:

=====

Figures - the inset text in most figures is much too small to read. Please make panels and/or font sizes bigger.

pg 4: "we computed the resultant" -> "we computed the resultant, or mean vector across environments". (Some readers may not be familiar with the term "resultant", so it would be helpful to add some extra explanation).

pg 4: "provides an estimate of direction scatter which, in this experiment, was $\bar{d}_s = 0.026$ ". In radians or degrees? (Should have units).

pg 4: Section: "Response directions are approximately invariant"

This is a very cool result. However, I can't help but wonder: even if the scatter in angles is very small, is there any systematic shift in the angles that is worth reporting? e.g., does the angle shift perhaps towards (or away from) the axis defined by preferred stimuli that were over or under-represented in the probability distribution? It seems this would be worth reporting even if the primary result is that the angle change is very small.

pg 4: Section: "The power law predicts response magnitudes in novel environments."

This is a fabulous result! My only comment is that I think the authors should add a caveat related to the rough order of magnitude over which this result was tested. So for example, can we be certain that $P_A(s) / P_B(s) = 10$ or 20 , the power law still holds?

Fig 2 caption says only "Testing for population homeostasis" for column (vii). This needs a bit more explanation. it's not even clear what the dotted and solid lines are. (And legend should point out that these curves should be flat if there were homeostasis).

Fig 3 caption: "optimal von Mises kernel".

pg 6: "Using the optimal κ largely restores the power law relationship "

What is optimal about it? Need a bit more explanation / motivation here.

Similarly, I'm not sure what "adjusted R-squared" refers to in either the text or the fig caption. (This is R^2 for what? And how was it adjusted?)

Pg 6:

I found it slightly odd / confusing that there is a section titled "The rules of adaptation extend to natural distributions" followed by a section titled "The rules of adaptation extend to complex stimulus sets". The first refers to experiments with static natural images, while the second refers to experiments with natural movies. This is not at all evident from the titles, however. (Both are natural and both are complex!) Suggest giving more descriptive headings, eg ("Natural images" and "Natural movies"?)

Pg 6-7: for the experiments with movie clips, how were the population response vectors defined? (ie. was it still a vector spike counts across neurons, resulting in a length N vector for a population of N neurons, or was it derived from an $N \times T$ matrix, for N neurons by T time bins?) Should clarify this point in the main text. It seems unfair to compare cosine distances between response vectors if we have increase the dimensionality of the response vector. (In higher dimensions, it's unsurprising that the response vectors are more orthogonal). Or if we haven't increased dimensionality, then what is the justification for throwing away temporal information by considering only a single time bin?

Fig 6 caption: panel D needs more explanation.

pg 9: "if we take expectations on both sides of the "surprise" equation" - presumably this is expectation over $P_x(s)$, the distribution of stimuli, but you should say this explicitly.

pg 9: "This means that adaptation enables the cortex to adjust the metabolic cost of its representation to the predictability of the environment."

There is something slightly odd about the argument that expected log firing rate is a measure of metabolic activity. (Is there some justification for this claim on metabolic grounds)?

If we zoom out slightly, consider that if Beta were equal to -1, we would have so-called "perfect adaptation", where the mean activity is constant across all environments. However, Beta = -1 would mean that $E[\log R_x(s)]$ is even higher for high-entropy environments than it is with the observed values of Beta ~ -0.4 . But in the case of Beta=-1, if we say that expected log firing rate is a measure of metabolic activity, we are forced into saying that high-entropy environments exact a higher metabolic cost even though the average firing rate did not change, which seems a bit bizarre.

This is a narrow comment related to the main criticism above about the lack of motivation for the information-theoretic analyses, so the authors only need to address this point if they wish to keep these analyses in the paper.

Reviewer #2 (Remarks to the Author):

Recent experience robustly shapes sensory processing. Yet, lawful dependencies of the effects of a given stimulus set has been hard to determine because of seemingly diverse effects on response amplitude that depend on the heterogenous tuning preferences of the population. Tring et al., take a clever perspective on the problem by addressing the effect of adaptation on the population code. They reveal a remarkably fixed relationship between the magnitude of change in the population vector in response to any given stimulus following adaptation and its relative frequency in the set of stimuli. This relationship is so stereotyped that they are able to predict the magnitude of adaptation in response to a completely novel stimulus distribution. They extend this principle to demonstrate that it holds with both more natural distributions of stimulus statistics and in response to short natural movies. Overall, this is an theoretically important contribution that brings significant clarity our understanding of the impact of adaptation on sensory processing and efficient coding. For one, this provides the exponent of the

relationship as a useful measure of adaptation that can be compared across stimulus features, brain areas, species, etc. I only have a few minor suggestions for improving the clarity of the manuscript.

Science:

1. The authors make the very interesting observation that adaptation does not enable complete homeostasis across the population as previously observed by Benucci and colleagues. Indeed, as the authors point out, this would require a much larger exponent than they typically observe.

a. It would be useful to understand the degree to which some homeostasis is achieved- could the authors provide a plot like Figure 2a(vii) which overlays the distributions from the control conditions to visualize how much flatter/smaller the adapted ones become? A quantification of homeostasis, and its increase, would also be helpful.

b. The authors demonstrate that the difference in stimulus statistics cannot fully explain the difference in homeostasis observed. However, they point out that in order to assess this, they need to add a filter to the stimulus to account for the tuning width of V1 neurons in the mouse. Yet, orientation tuning in the cat is considerably sharper than in the mouse, and therefore should not smooth the distribution as strongly. Could this explain the differences in homeostasis across species?

2. It is interesting that the exponents measured for the natural movies were significantly smaller than for stimulus sets composed of varying distributions of oriented gratings. The authors should discuss what features determine the slope of the power relationship, and how this impacts their ability to reliably predict across environments.

3. The recovery from adaptation is asymmetrical for facilitation and suppression (~2x difference in tau). Does this suggest that they are due to different processes? These rates are also quite a bit faster than reported by Fritsche et al. 2022. Is this related to the incorporation of multiple stimulus phases?

Formatting:

1. This is a very mathematical approach and therefore including the equations in the Results section is important. However, many readers may not be fluent in the notations and would benefit from a more intuitive description of the metrics being calculated. For instance, on lines 92-93 a verbal explanation of the normalization performed to calculate \hat{r} and the quantity defined by \bar{r} , would be helpful. Similarly, some of the language is challenging- for instance, the first clause of the sentence on line 75 doesn't obviously have a verb (or a subject?).

2. The introduction is unorthodox for this journal. The authors should consider spending more time setting up the importance of understanding the relationship between adaptation magnitude and environment.

3. The Figures are way too small. The axis labels in Figure 2 are 3 pt font. I appreciate that the authors want to fit all of these plots into a single figure, and I certainly appreciate seeing the data for multiple independent experiments, but the figures need to be legible on a piece of paper. Perhaps moving some of the additional experiments to the supplement and providing a summary figure that demonstrates the consistency across experiments would be a reasonable compromise.

4. It's not obvious in Figure 7d that the tuning curves near to the adapter are skewed away from it- the yellow and red lines look fairly symmetrical to me.

5. There are a couple of typos:

Line 14- "independently sampling form its distribution"

Line 233- "the responses magnitudes"

Reviewer #3 (Remarks to the Author):

Summary and Contributions

The authors consider population-level adaptation phenomena, which are relatively understudied in the field of neural adaptation. This paper adopts a probabilistic stimulus adaptation framework, generalizing the paradigm in Benucci et al. (2013), as opposed to the more commonly used adapter top-up paradigm to study population adaptation in early vision.

One main result is that the ratio of mean population responses, measured via 2-photon calcium imaging in layers 2-3 of mouse V1 (as opposed to cat V1 in the Benucci study), follow a power law relationship dependent on the ratio of corresponding stimulus probabilities. Notably, the authors argue convincingly that adaptive homeostasis effects previously reported by Benucci and colleagues cannot explain all the response changes they observed.

This a well done and timely study that presents findings which challenge our current understanding and build new normative motivations for adaptive population coding. It is for the most part clearly written and well framed, though I found the second results subsection “Response directions are approximately invariant”, confusing. Some work on the writing would be welcome here.

As the authors have demonstrated, using a well-defined probabilistic stimulus enables more rigorous analyses, and consideration of efficient coding principles, rooted in statistics and information theory. This is a significant advance and I believe more studies should adopt this kind of paradigm. The authors are also clear about the possible limitations, e.g. discrepancies between the experimental setups of their study and Benucci et al. (different animal models, spikes vs 2P imaging, etc)

Weaknesses and suggestions

The authors missed an opportunity to explore adaptive changes in covariance. They are aware of this, mentioning it in their discussion, but a finding of signal covariance redundancy reduction would make their appeal to adaptive coding efficiency stronger (e.g. Barlow & Foldiak, 1989; Mlynarski and Hermundstad, 2021). It would be interesting to see how the covariance (analogous to Benucci’s case, using relative/scaled covariance) changes according to the stimulus probabilities.

A few sentences elaborating on why the authors’ calcium imaging methodology (as opposed to spikes in the Benucci paper) may contribute to differences between their findings would be useful.

It’s possible that the homeostasis hypothesis refuted by the authors, involving statistics essentially being exactly matched before and after adaptation, is too strict a constraint. One could imagine a softer form of homeostasis, in which adaptation tilts the population response toward the statistics of the previous context, but does not necessarily match it. An analysis showing simulated population responses with/without adapting to the new stimulus probability distribution might reveal such an effect. Take Fig 2a-vii with contexts A and C for example, which seems to be most similar to the Benucci experiments involving biased vs unbiased distributions. It would be helpful to show an analysis to better understand what the neural responses would look like without adapting to the ensemble. This would provide more direct comparison to Fig. 2 in the Benucci paper (and Fig 3, if they end up doing a signal covariance analysis), and would reveal whether or not a “soft” homeostasis effect is occurring.

The paper would benefit from a more elaborate discussion of their conjectured synaptic depression mechanism.

Reviewer #4 (Remarks to the Author):

Tring et al study effects of adaptation on population level in the mouse visual cortex using two photon calcium imaging. In particular, they study adaptation to stimulus probabilities by using stimuli sets with differing distributions. They show two main results, one that amplitude of populations responses in different environments can be explained power law between ratio of amplitudes and ratio of stimulus probabilities. The power coefficient is not -1 however, as would be perfect homeostasis, but between -0.15 to -0.4. A previous study had arrived at a different conclusion, but it was using a different species and different techniques. The second main result is to show that direction of the population response vector remains invariant in changing environments. These results offer useful good insights into studying mechanisms of adaptation and predicting responses in novel environments and will be of value to the field. Below are some questions and comments to improve the manuscript, I support its publication pending author responses.

1. How well is the power law preserved on individual trials of the stimulus? Is there any change in beta if initial vs late repeats are used from a given environment to estimate beta?
2. Adaptation is known to occur over multiple timescales (see Fairhall et al, 2000, Ulanovsky et al 2004, Latimer et al 2019). Yet, the current study finds that stimuli more than 2 seconds into the past do not modulate the responses. Can the authors comment on why they don't observe adaptation on longer time scales? It could be dependent on the stimulus presentation frequency maybe as only one frequency was used to obtain figure 6.
3. The estimated beta exhibits some variability. Can the authors comment on how much of this variability is related to stimulus environments. For example, if beta were estimated for each pair of environments, would it correlate to cosine distance between the 2 environments?
4. Indicate the number of orientations used for each environment and how many times each stimulus was repeated in an environment.
5. Indicate that recording is from mouse brain in the main text.
6. Fix the labels of stimulus environments, they don't match the text for figure 2A(i), A is the uniform distribution and C is centered at an orientation.
7. Include population response profiles like in figure 2A(ii) for supplementary figure 1A and B. How many cells were discarded using the selectivity criterion?
8. It will be helpful to see how the results are dependent on spike inference algorithm. Since the inference process is noisy and only explains variance observed in spike rate partially, it would be useful to have a supplementary figure showing how the ratio of responses depends on stimulus probability ratios using direct calcium responses.
9. How is cosine distance calculated? The distribution mean is around 0, does it mean the direction vectors in different environments are orthogonal to each other?
10. In legend for supplementary figure 1 "The asymptotic cosine distance between response vectors between two angles is lower when we include all neurons", don't you mean higher?

11. In methods, under analysis of population homeostasis, second last line of the paragraph, the equation for power law with beta -1, numerator subscript should be P_y not x .

12. The procedure to create environments from natural statistics needs more details. An orientation distribution was obtained for each image patch. Did the 150 distributions correspond to 150 image patches? For the set with movie presentation, what was the scheme of stimulus presentation? A schematic like figure 1b would help explain this. Were the 18 clips repeated? How was the environment defined for movie stimulus? From figure 5b(i) it appears that each environment was composed of multiple clips with differing probability, in which case, it's not clear what about this distribution was similar to natural orientation distributions? Was orientation distribution obtained for each movie frame? How was similarity assessed between natural orientation distribution and movie environment distributions?

13. How was scatter between population vectors for individual orientation and response to each orientation $R(\theta)$ obtained for movie clips? Were gratings presented to match the movie orientation distribution? If so, this also needs to be specified. Was the stimulus used for figures 2b,c,d(ii-vi) gratings or movie clips? Because if movie clips are presented, each frame must be composed of multiple orientations and it's not clear how $R(\theta)$ and direction vector for each orientation were obtained in that case.

We thank the reviewers for their constructive suggestions and criticisms.

Overall, the manuscript was well received. Reviewer #1 remarked: "This is a stunning set of results, representing an experimental and theoretical tour de force [...] It is one of the best papers I've read all year, and I'm very excited to see it in print." Reviewer #2, noted that "this is a theoretically important contribution that brings significant clarity our understanding of the impact of adaptation on sensory processing and efficient coding." Reviewer 3 wrote: "This a well done and timely study that presents findings which challenge our current understanding and build new normative motivations for adaptive population coding." And Reviewer #4 remarked: "These results offer useful good insights into studying mechanisms of adaptation and predicting responses in novel environments and will be of value to the field."

In addition to their overall positive assessment, the reviewers raised some important questions and offered ways to strengthen the manuscript. In this revised version, we have adopted many of the reviewers' suggestions for improvement. To summarize, the major changes include:

- Following Reviewer's #1 recommendation, we have dropped the original Fig 6d.
- Following a comment by Reviewer #1, we have now redefined our measure of metabolic cost and offer a new analytic treatment to study the relationship between metabolic cost and the entropy of the environments in a new Figs 6d,e.
- We have added a new Fig 6f following a recommendation from both Reviewers #2 and #3 to provide a visual depiction of the degree by which different exponents provide different degrees of adaptation.
- We have added a new Fig 7f, to respond to Reviewer #2 comment about changes in the skewness of the tuning curves being not immediately obvious from simple, visual inspection.
- We have added Supp Fig 1c in response to Reviewer #4 request to provide a comparison of data using different deconvolution algorithms.
- In reply to a question by Reviewer #4, we added a new section to the methods to describe in more detail the stimuli used in the movie sequences.
- All figures have been modified to use larger fonts.
- The text has been extensively revised (with tracking) to respond to the reviewer's requests for clarification.

A point-by-point reply to their comments follows.

REVIEWER COMMENTS

Reviewer #1 (Remarks to the Author):

This paper presents a beautiful theoretical analysis of adaptation in neural population responses. Specifically, it shows that --- across a wide variety of stimulus environments --- the ratio between the amplitudes of the population responses to a given stimulus between two environments is a power law of the ratio between the probabilities of that stimulus in the two environments. A power of Beta = -1

would correspond to "perfect adaptation", where the total number of spikes (or "RMS spikes", depending on the definition of amplitude) elicited by each stimulus is constant across environments, while a power of $\beta = 0$ would correspond to "no adaptation", where the response amplitudes do not change across environments. Here the authors report a power of β in the range $[-0.4, -0.15]$, where specific value depends somewhat on the details of the stimulus set. Thus, real neural populations sit somewhere short of perfect adaptation, with the consequence (as the paper indeed shows) adaptation does NOT achieve population level homeostasis. Additionally, the authors report that the angle of the population response vector does *not* change meaningfully across adaptive states.

This is a stunning set of results, representing an experimental and theoretical tour de force. The paper is clearly written, and the figures are beautiful (with the exception of the small font size!). It is one of the best papers I've read all year, and I'm very excited to see it in print. I have only one major comment.

Thank you for this assessment.

Major comment:

=====

My only major criticism is that I am not convinced that the information theoretic analysis in Fig 6 (panel d) makes sense, or adds much of value to the paper.

It seems fine to point out that the power law relationship implies that the log amplitude increases as a function of "surprise". However, the logic and motivation of this section is otherwise a bit confusing.

Recall that we are focusing on only 1 dimension of the high dimensional population activity: the L1 or L2 norm of the population response vector. Presumably the stimulus identity itself is encoded very accurately by the angle of the response vector (which as the paper shows, does not vary with the environment). Thus we are NOT talking about the encoding of the stimulus itself. But then what kind of information are we talking about? Is it information about $P_x(S)$? (But isn't that something we could get more accurate information about by keeping track of the response vector angles?) Is it obvious that this is something the brain cares about or needs to encode? If the authors do indeed mean information about the adaptive state or environment, then it would be more useful to ask questions about how you might decode this information, rather than merely quantifying it with bits or KL divergence.

Fig 6D focuses on the KL divergence between two stimulus distributions, and showing that we can extract this from the expected log ratio of response amplitudes. However, this seems like a mathematical curiosity that follows straightforwardly from power law adaptation and the definition of KL divergence, and I'm not convinced that this is relevant or meaningful. Can the authors offer a more convincing argument for why this would matter? (Do they wish to argue that the brain needs to compute the KL divergence between different distributions over stimuli for some reason? Or is this quantity useful in some way for a computation the brain needs to perform?)

In its current form, this analysis feels like a bit of an afterthought, and as a result it seems to me to detract from rather than add to the main message. So I would suggest that the authors either motivate and explain the information-theoretic argument more fully and clearly or else feel free to remove it. (I think the paper stands quite nicely on its own even without it!)

We agree the KL divergence analysis is somewhat disconnected from the main line of the paper. We have accepted the reviewer's suggestion and have now removed this analysis from the manuscript.

=====
Detailed comments:
=====

Figures - the inset text in most figures is much too small to read. Please make panels and/or font sizes bigger.

Fixed. Font sizes are now larger in all figures.

pg 4: "we computed the resultant" -> "we computed the resultant, or mean vector across environments". (Some readers may not be familiar with the term "resultant", so it would be helpful to add some extra explanation).

Thank you, we have added this clarification.

pg 4: "provides an estimate of direction scatter which, in this experiment, was $\bar{d}_s = 0.026$ ". In radians or degrees? (Should have units).

In the sentence prior to this statement, we stipulate \bar{d}_s is the scatter of cosine distance values, which is, therefore, unitless.

pg 4: Section: "Response directions are approximately invariant"

This is a very cool result. However, I can't help but wonder: even if the scatter in angles is very small, is there any systematic shift in the angles that is worth reporting? e.g., does the angle shift perhaps towards (or away from) the axis defined by preferred stimuli that were over or under-represented in the probability distribution? It seems this would be worth reporting even if the primary result is that the angle change is very small.

Yes, there is a systematic shift. Indeed, this is the topic of a separate manuscript currently under preparation where we describe these systematic shifts along with changes in discriminability. Unfortunately, the inclusion of these results is not possible within the scope of this paper, where we decided to focus largely on the description of the power law.

pg 4: Section: "The power law predicts response magnitudes in novel environments."

This is a fabulous result! My only comment is that I think the authors should add a caveat related to the rough order of magnitude over which this result was tested. So for example, can we be certain that $P_A(s) / P_B(s) = 10$ or 20 , the power law still holds?

It is a very good question. Unfortunately, the total experimental time limits the range of reasonable probability ratios we can explore. This is because reducing the minimum probability of a stimulus determines the minimum, expected number of trials we can obtain for a stimulus. We aimed to have at

least 50 trials per stimulus. We have added a statement in the Discussion clarifying this limitation of the data.

Fig 2 caption says only "Testing for population homeostasis" for column (vii). This needs a bit more explanation. it's not even clear what the dotted and solid lines are. (And legend should point out that these curves should be flat if there were homeostasis).

Thank you. We have now added a more detailed explanation to the caption.

Fig 3 caption: "optimal von Mises kernel".

pg 6: "Using the optimal κ largely restores the power law relationship "

What is optimal about it? Need a bit more explanation / motivation here.

Similarly, I'm not sure what "adjusted R-squared" refers to in either the text or the fig caption. (This is R^2 for what? And how was it adjusted?)

The power law is characterized by linear relationship in log-log coordinates (of probability ratios vs magnitude ratios). In page 3, we state we use the (adjusted) R^2 is a goodness-of-fit measure of the **linear fit**. As noted in the caption of Fig 2, the optimal linear fits are represented by the solid lines in the graphs (for example, the solid lines in Fig 2 column (iii)).

In the experiments of Fig 3 (peaked distributions), we observed that the linear relationship fits the data better if we smooth the probabilities of the environment with a von-Mises kernel. In the second paragraph of page 6, we explain that the optimal smoothing is defined as the one that yields the best linear fit (the one which attains the maximum R-squared statistic).

Regarding the calculation of the "adjusted" R^2 –

In general, the adjusted R^2 is computed as: $R^2 = 1 - \frac{SS_{res}/df_{res}}{SS_{tot}/df_{tot}}$. The goal of the adjustment is to compensate for the number of variables in a linear model (which is useful when comparing goodness of fit between models with different number of parameters). Here, SS_{res} is the sum of squares of the residuals after the model fit, while SS_{tot} is the total sum of squares (proportional to the variance of the data). In our case, the degree of freedom of the residuals is $df_{res} = n - 1$, where n is the sample size (as our model has only one parameter), and $df_{tot} = n - 1$, as the model in this case is just the mean value of the data. Thus, for the case of a linear model with no intercept we have $R^2 = 1 - \frac{SS_{res}}{SS_{tot}}$, which is the same as the standard coefficient of determination R^2 . Thus, we have removed "adjusted" from this version of the manuscript as the two are the same in our case and we are not performing any model comparisons.

Pg 6:

I found it slightly odd / confusing that there is a section titled "The rules of adaptation extend to natural distributions" followed by a section titled "The rules of adaptation extend to complex stimulus sets". The first refers to experiments with static natural images, while the second refers to experiments with natural movies. This is not at all evident from the titles, however. (Both are natural and both are complex!) Suggest giving more descriptive headings, eg ("Natural images" and "Natural movies"?)

This was confusing. We have modified the titles of these sections accordingly.

Pg 6-7: for the experiments with movie clips, how were the population response vectors defined? (ie. was it still a vector spike counts across neurons, resulting in a length N vector for a population of N neurons, or was it derived from an NxT matrix, for N neurons by T time bins?) Should clarify this point in the main text. It seems unfair to compare cosine distances between response vectors if we have increase the dimensionality of the response vector. (In higher dimensions, it's unsurprising that the response vectors are more orthogonal). Or if we haven't increased dimensionality, then what is the justification for throwing away temporal information by considering only a single time bin?

Thank you. The processing was the same as in the case with gratings: the optimal time delay was computed as the one leading to the peak response magnitude after the onset of a stimulus. The responses were analyzed at this optimal time. We have now clarified this point in the Methods. We chose to keep the processing of the data the same across different stimuli sets so we could ascribe any observed changes to the different stimulus ensembles.

Fig 6 caption: panel D needs more explanation.

Following the reviewer's recommendation, we have now removed this panel.

pg 9: "if we take expectations on both sides of the "surprise" equation" - presumably this is expectation over $P_x(s)$, the distribution of stimuli, but you should say this explicitly.

Correct, we now state this explicitly.

pg 9: "This means that adaptation enables the cortex to adjust the metabolic cost of its representation to the predictability of the environment."

There is something slightly odd about the argument that expected log firing rate is a measure of metabolic activity. (Is there some justification for this claim on metabolic grounds)?

If we zoom out slightly, consider that if Beta were equal to -1, we would have so-called "perfect adaptation", where the mean activity is constant across all environments. However, Beta = -1 would mean that $E[\log R_x(s)]$ is even higher for high-entropy environments than it is with the observed values of Beta ~ -0.4 . But in the case of Beta=-1, if we say that expected log firing rate is a measure of metabolic activity, we are forced into saying that high-entropy environments exact a higher metabolic cost even though the average firing rate did not change, which seems a bit bizarre.

This is a narrow comment related to the main criticism above about the lack of motivation for the information-theoretic analyses, so the authors only need to address this point if they wish to keep these analyses in the paper.

We agree that defining metabolic cost as $E\{\log r_X(s_i)\}$ may be unfounded and lead to some logical inconsistencies. In response, we have now replaced this analysis by defining metabolic cost as the expected value of the population magnitude $E\{r_X(s_i)\}$. We analyze the relationship between metabolic cost and entropy for the class of von Mises distributions to demonstrate that metabolic cost decreases with entropy under the range of exponents observed (new Figs 6e,e). Moreover, as the reviewer correctly pointed out, metabolic cost is constant under perfect adaptation or no adaptation at all.

Reviewer #2 (Remarks to the Author):

Recent experience robustly shapes sensory processing. Yet, lawful dependencies of the effects of a given stimulus set has been hard to determine because of seemingly diverse effects on response amplitude that depend on the heterogenous tuning preferences of the population. Tring et al., take a clever perspective on the problem by addressing the effect of adaptation on the population code. They reveal a remarkably fixed relationship between the magnitude of change in the population vector in response to any given stimulus following adaptation and its relative frequency in the set of stimuli. This relationship is so stereotyped that they are able to predict the magnitude of adaptation in response to a completely novel stimulus distribution. They extend this principle to demonstrate that it holds with both more natural distributions of stimulus statistics and in response to short natural movies. Overall, this is an theoretically important contribution that brings significant clarity our understanding of the impact of adaptation on sensory processing and efficient coding. For one, this provides the exponent of the relationship as a useful measure of adaptation that can be compared across stimulus features, brain areas, species, etc. I only have a few minor suggestions for improving the clarity of the manuscript.

Science:

1. The authors make the very interesting observation that adaptation does not enable complete homeostasis across the population as previously observed by Benucci and colleagues. Indeed, as the

authors point out, this would require a much larger exponent than they typically observe.

a. It would be useful to understand the degree to which some homeostasis is achieved- could the authors provide a plot like Figure 2a(vii) which overlays the distributions from the control conditions to visualize how much flatter/smaller the adapted ones become? A quantification of homeostasis, and its increase, would also be helpful.

Following the reviewer's recommendation, we added a new Fig 6f to visually depict the effect of different powers on the population amplitude for an orientation distribution following a standard von Mises distribution. This case is simple to understand because, for a von Mises, we have $p(\theta) \sim \exp(\kappa \cos(\theta))$, which leads to the function $\Phi(\theta) \cong p(\theta)r(\theta) = p(\theta)p(\theta)^\beta = p(\theta)^{(\beta+1)} \sim \exp(\kappa(1 + \beta)\cos(\theta))$. In other words, for a von Mises distribution and a power law, the mean population responses as a function of orientation are also von Mises with a concentration parameter $\kappa(1 + \beta)$.

b. The authors demonstrate that the difference in stimulus statistics cannot fully explain the difference in homeostasis observed. However, they point out that in order to assess this, they need to add a filter to the stimulus to account for the tuning width of V1 neurons in the mouse. Yet, orientation tuning in the cat is considerably sharper than in the mouse, and therefore should not smooth the distribution as strongly. Could this explain the differences in homeostasis across species?

Perhaps. As the reviewer knows, some cells in mouse V1 are as well tuned as those in cats (for example, Niell and Stryker, 2008) and we selected cells with good tuning in our analyses, so it we think it is difficult to predict what would happen if we were to conduct the same experiment in cats. We agree with the reviewer that exploring potential interspecies differences is important. We are now collaborating with the Huk Lab at UCLA to perform these experiments in marmoset V1 to perform such comparative studies.

2. It is interesting that the exponents measured for the natural movies were significantly smaller than for stimulus sets composed of varying distributions of oriented gratings. The authors should discuss what features determine the slope of the power relationship, and how this impacts their ability to reliably predict across environments.

This is a good question. We speculate this may be due to the degree to which different stimuli generate activity in the same set of neurons. Pairs of sinusoidal gratings differing by 10 deg in orientation probably stimulate similar sets of neurons. This can be inferred by the fact that the average cosine distance between the population activity in this case is approximately 0.15 (Fig 2 (v)). In contrast, the cosine distance between any two pairs of movie clips was larger than 0.5, meaning that different movies generated very different patterns and, therefore, would adapt independent set of neurons. Future experiments could explore if there is a relationship between the exponent and the pairwise cosine distance a stimulus set evokes in single populations. We have now mentioned this possibility in the Discussion.

3. The recovery from adaptation is asymmetrical for facilitation and suppression (~2x difference in tau).

Does this suggest that they are due to different processes? These rates are also quite a bit faster than reported by Fritsche et al. 2022. Is this related to the incorporation of multiple stimulus phases?

Although the dynamics are roughly in agreement, as the reviewer notes there are some important differences between the studies that may lead to somewhat different results. One is the difference in the timing of the stimuli and the inclusion of different phases (which are averaged in our analyses). Their analyses are based on the responses of single cells, while our analyses are based on population magnitudes. Perhaps details about population selection could result in differences. Finally, the recording modalities differed: we used two photon imaging data while Fritsche et al relied on Neuropixels recordings.

Formatting:

1. This is a very mathematical approach and therefore including the equations in the Results section is important. However, many readers may not be fluent in the notations and would benefit from a more intuitive description of the metrics being calculated. For instance, on lines 92-93 a verbal explanation of the normalization performed to calculate \hat{r} and the quantity defined by \bar{r} , would be helpful. Similarly, some of the language is challenging- for instance, the first clause of the sentence on line 75 doesn't obviously have a verb (or a subject?).

Thank you. In this revised version, we have added additional descriptions surrounding the formulas in words to clarify the concepts.

2. The introduction is unorthodox for this journal. The authors should consider spending more time setting up the importance of understanding the relationship between adaptation magnitude and environment.

Thank you. We think the paragraph in the revised Introduction section and the first paragraph of the Results provide a succinct summary which clearly describes the importance and goals of the study.

3. The Figures are way too small. The axis labels in Figure 2 are 3 pt font. I appreciate that the authors want to fit all of these plots into a single figure, and I certainly appreciate seeing the data for multiple independent experiments, but the figures need to be legible on a piece of paper. Perhaps moving some of the additional experiments to the supplement and providing a summary figure that demonstrates the consistency across experiments would be a reasonable compromise.

Apologies for the small font sizes, we have now increased the font sizes in all figures.

4. It's not obvious in Figure 7d that the tuning curves near to the adapter are skewed away from it- the yellow and red lines look fairly symmetrical to me.

Instead of relying exclusively on visual inspection, we now added a new panel showing how circular skewness changes as a function of the cell's preferred orientation (**Fig 7f**). We also highly curves where the difference may be more obvious.

5. There are a couple of typos:

Line 14- "independently sampling form its distribution"

Line 233- "the responses magnitudes"

Thank you. This has now been fixed.

Reviewer #3 (Remarks to the Author):

Summary and Contributions

The authors consider population-level adaptation phenomena, which are relatively understudied in the field of neural adaptation. This paper adopts a probabilistic stimulus adaptation framework, generalizing the paradigm in Benucci et al. (2013), as opposed to the more commonly used adapter top-up paradigm to study population adaptation in early vision.

One main result is that the ratio of mean population responses, measured via 2-photon calcium imaging in layers 2-3 of mouse V1 (as opposed to cat V1 in the Benucci study), follow a power law relationship dependent on the ratio of corresponding stimulus probabilities. Notably, the authors argue convincingly that adaptive homeostasis effects previously reported by Benucci and colleagues cannot explain all the response changes they observed.

This a well done and timely study that presents findings which challenge our current understanding and build new normative motivations for adaptive population coding. It is for the most part clearly written and well framed, though I found the second results subsection "Response directions are approximately invariant", confusing. Some work on the writing would be welcome here.

As the authors have demonstrated, using a well-defined probabilistic stimulus enables more rigorous analyses, and consideration of efficient coding principles, rooted in statistics and information theory. This is a significant advance and I believe more studies should adopt this kind of paradigm. The authors are also clear about the possible limitations, e.g. discrepancies between the experimental setups of their study and Benucci et al. (different animal models, spikes vs 2P imaging, etc)

Weaknesses and suggestions

The authors missed an opportunity to explore adaptive changes in covariance. They are aware of this, mentioning it in their discussion, but a finding of signal covariance redundancy reduction would make their appeal to adaptive coding efficiency stronger (e.g. Barlow & Foldiak, 1989; Mlynarski and Hermundstad, 2021). It would be interesting to see how the covariance (analogous to Benucci's case, using relative/scaled covariance) changes according to the stimulus probabilities.

We agree these are important questions. As we note above and in the Discussion, a separate manuscript currently under preparation will describe changes in population direction, response co-

variance, and discriminability during adaptation. Our goal in the present manuscript was to focus on the description of the power law property.

A few sentences elaborating on why the authors' calcium imaging methodology (as opposed to spikes in the Benucci paper) may contribute to differences between their findings would be useful.

We have elaborated on the possibility that calcium imaging nonlinearities may be present during imaging (see Nauhaus et al (2012)) and how any such effects may be revealed by comparing 2p and electrophysiological recordings using the same visual stimulation methods.

It's possible that the homeostasis hypothesis refuted by the authors, involving statistics essentially being exactly matched before and after adaptation, is too strict a constraint. One could imagine a softer form of homeostasis, in which adaptation tilts the population response toward the statistics of the previous context, but does not necessarily match it. An analysis showing simulated population responses with/without adapting to the new stimulus probability distribution might reveal such an effect. Take Fig 2a-vii with contexts A and C for example, which seems to be most similar to the Benucci experiments involving biased vs unbiased distributions. It would be helpful to show an analysis to better understand what the neural responses would look like without adapting to the ensemble. This would provide more direct comparison to Fig. 2 in the Benucci paper (and Fig 3, if they end up doing a signal covariance analysis), and would reveal whether or not a "soft" homeostasis effect is occurring.

Indeed, as the reviewer points out, an intermediate value of beta between -1 and zero would represent a regime of "soft" homeostasis. We clarify this in the text, and now we offer a new Fig 6f, to clarify the degree to which magnitudes are equalized for different values of the exponent.

The paper would benefit from a more elaborate discussion of their conjectured synaptic depression mechanism.

We now mention the work of Nelson and colleagues on thalamocortical depression and conjecture if a simple, feedforward mechanism may explain the results. We have some preliminary data showing that such models may work to explain the data, but we feel it is premature for us to go beyond a simple conjecture.

Reviewer #4 (Remarks to the Author):

Tring et al study effects of adaptation on population level in the mouse visual cortex using two photon calcium imaging. In particular, they study adaptation to stimulus probabilities by using stimuli sets with differing distributions. They show two main results, one that amplitude of populations responses in different environments can be explained power law between ratio of amplitudes and ratio of stimulus probabilities. The power coefficient is not -1 however, as would be perfect homeostasis, but between -0.15 to -0.4. A previous study had arrived at a different conclusion, but it was using a different species and different techniques. The second main result is to show that direction of the population response vector remains invariant in changing environments. These results offer useful good insights into

studying mechanisms of adaptation and predicting responses in novel environments and will be of value to the field. Below are some questions and comments to improve the manuscript, I support its publication pending author responses.

1. How well is the power law preserved on individual trials of the stimulus? Is there any change in beta if initial vs late repeats are used from a given environment to estimate beta?

If we use half the data at the beginning or at the end of the presentation of environments to estimate beta, we do not observe any systematic differences. It is worth noting there that there are other non-stationarities in the experiment that are expected to substantially influence the magnitude of responses during one session and may dwarf any other effects. For example, fluctuations in locomotion or alertness are a source of strong modulation of population responses. In the present analyses, we randomly interleaved environments in the hope of averaging over them. We have kept locomotion and eye movement data, but we have not yet made a systematic effort to study how their variability across trials may affect our estimation of the exponent.

2. Adaptation is known to occur over multiple timescales (see Fairhall et al, 2000, Ulanovsky et al 2004, Latimer et al 2019). Yet, the current study finds that stimuli more than 2 seconds into the past do not modulate the responses. Can the authors comment on why they don't observe adaptation on longer time scales? It could be dependent on the stimulus presentation frequency maybe as only one frequency was used to obtain figure 6.

It is a great question. Unfortunately, as the reviewer points out, studying adaptation at multiple time scales would require us to perform the same experiment at different stimulus presentation rates and see if the temporal responses scale accordingly. The present study was limited to just one, but future studies could certainly address this question.

3. The estimated beta exhibits some variability. Can the authors comment on how much of this variability is related to stimulus environments. For example, if beta were estimated for each pair of environments, would it correlate to cosine distance between the 2 environments?

We suspect the reviewer is right: the farther apart the environments are, the better of an estimate we can get for the slope. The reason is that environments that are very similar will result in a range of probability ratios between the environments that is smaller than a pair of very different environments. Fitting a line with a larger range of values is likely to yield a better estimate of the slope. This intuition was also part of the reason we required samples from natural environments to have a cosine distance at least 0.25 apart. The raw data to be shared in Figshare contains the empirical distribution of the orientations allowing for the exploration of this question.

4. Indicate the number of orientations used for each environment and how many times each stimulus was repeated in an environment.

Thank you. This information has now been added under the "Visual stimulation" section in the Methods.

5. Indicate that recording is from mouse brain in the main text.

Done! (in the first paragraph).

6. Fix the labels of stimulus environments, they don't match the text for figure 2A(i), A is the uniform distribution and C is centered at an orientation.

Thank you. We fixed the text (the figure was correct).

7. Include population response profiles like in figure 2A(ii) for supplementary figure 1A and B. How many cells were discarded using the selectivity criterion?

Cells that are not well tuned do not have a well-defined preferred orientation, making their inclusion in the response profiles not meaningful or informative (as cells are ordered by their preferred orientation). To answer the question, on average, about 30% of ROIs pass the selection criteria (see selected and total cells in Supplementary Table 1), resulting in experiments with an average population size of ~400 neurons. The Figshare repository we provide raw data from all ROIs, making it possible for others to reanalyze the data using other selection criteria.

8. It will be helpful to see how the results are dependent on spike inference algorithm. Since the inference process is noisy and only explains variance observed in spike rate partially, it would be useful to have a supplementary figure showing how the ratio of responses depends on stimulus probability ratios using direct calcium responses.

Following the reviewer's suggestion, have added Supp Fig 1c to illustrate the robustness of the power law to different deconvolution algorithms. We are currently conducting Neuropixels recordings to compare how the results depend on recording modality.

9. How is cosine distance calculated? The distribution mean is around 0, does it mean the direction vectors in different environments are orthogonal to each other?

Given two population vectors, the cosine distance is defined as one minus the cosine of the angle between them. Thus, the vectors are orthogonal when the distance between them is one. We are now stating this in the text for clarification. We are not entirely sure what distribution the reviewer is referring to.

10. In legend for supplementary figure 1 "The asymptotic cosine distance between response vectors between two angles is lower when we include all neurons", don't you mean higher?

No, we mean lower. We clarified this paragraph. As untuned cells are included in the analysis, many of them simply add noise to the representation and reduces the pairwise distance between population vectors.

11. In methods, under analysis of population homeostasis, second last line of the paragraph, the equation for power law with beta -1, numerator subscript should be P_y not x.

Indeed. Thank you for catching this. It has been fixed.

12. The procedure to create environments from natural statistics needs more details. An orientation distribution was obtained for each image patch. Did the 150 distributions correspond to 150 image patches? For the set with movie presentation, what was the scheme of stimulus presentation? A schematic like figure 1b would help explain this. Were the 18 clips repeated? How was the environment defined for movie stimulus? From figure 5b(i) it appears that each environment was composed of multiple clips with differing probability, in which case, its not clear what about this distribution was similar to natural orientation distributions? Was orientation distribution obtained for each movie frame? How was similarity assessed between natural orientation distribution and movie environment distributions?

Yes, each image resulted in one stimulus distribution. In total, we had 150 possible distributions to choose from. We clarified this in the text.

Apologies for the confusion created by an incomplete description of the movie clip stimuli. We have now added a new section to the methods to add the missing details. In response to your question, no measure of similarity is computed between natural movies to order them. Instead, movies are randomly assigned an ID. Thus, there is no relationship at all between the similarity of the movies and the similarity of their IDs.

13. How was scatter between population vectors for individual orientation and response to each orientation $R(\theta)$ obtained for movie clips? Were gratings presented to match the movie orientation distribution? If so, this also needs to be specified. Was the stimulus used for figures 2b,c,d(ii-vi) gratings or movie clips? Because if movie clips are presented, each frame must be composed of multiple orientations and it's not clear how $R(\theta)$ and direction vector for each orientation were obtained in that case.

The distribution of scatter for the movies can be calculated for individual elements of the stimuli set in the same way as they are for gratings (see description in page 3). Of course, what cannot be computed for the movies is the equivalent visual angle. Finally, to clarify, Fig 2 shows results for grating stimuli only.

REVIEWERS' COMMENTS

Reviewer #1 (Remarks to the Author):

I thank the authors for their detailed replies to my comments. The revised paper is substantially stronger, and I believe it is now ready for publication. Congratulations!

Reviewer #2 (Remarks to the Author):

The authors have addressed all of my concerns. Congrats on a beautiful manuscript.

Reviewer #3 (Remarks to the Author):

The authors have responded well and constructively to my comments and, as far as I can judge, to those of the other reviewers. I have no further comments. I think this is an outstanding piece of work that will significantly enhance our understanding.

Reviewer #4 (Remarks to the Author):

Tring et al study the relation between populations responses upon adaptation in different environments. They find the difference in responses amplitudes can be explained by a power law between the ratio of amplitudes and ratio of stimulus probabilities, with exponent smaller than 1. They also observe that direction of population responses vectors remains invariant across environments. The work presents important new insights into adaptation and how it affects sensory processing. The revisions have clarified my concerns. I have only one minor suggestion to add:

1. At the beginning of figure 2, showing some example cell responses (with a few different preferred orientations) under the three stimulus environments will help the reader to visualize what amplitude differences the paper is talking about. The tuning curves are provided in column ii, but they are average over environments and I think it would be useful to clearly depict how the amplitudes change using some example cell responses.

We thank the reviewers for their comments. Our replies follow.

REVIEWERS' COMMENTS

Reviewer #1 (Remarks to the Author):

I thank the authors for their detailed replies to my comments. The revised paper is substantially stronger, and I believe it is now ready for publication. Congratulations!

Thank you. We are glad the reviewer's questions were addressed satisfactorily.

Reviewer #2 (Remarks to the Author):

The authors have addressed all of my concerns. Congrats on a beautiful manuscript.

We are pleased the concerns have been addressed. Thank you.

Reviewer #3 (Remarks to the Author):

The authors have responded well and constructively to my comments and, as far as I can judge, to those of the other reviewers. I have no further comments. I think this is an outstanding piece of work that will significantly enhance our understanding.

Thank you for this positive assessment of our response to the review.

Reviewer #4 (Remarks to the Author):

Tring et al study the relation between populations responses upon adaptation in different environments. They find the difference in responses amplitudes can be explained by a power law between the ratio of amplitudes and ratio of stimulus probabilities, with exponent smaller than 1. They also observe that direction of population responses vectors remains invariant across environments. The work presents important new insights into adaptation and how it affects sensory processing. The revisions have clarified my concerns.

Thank you.

I have only one minor suggestion to add:

1. At the beginning of figure 2, showing some example cell responses (with a few

different preferred orientations) under the three stimulus environments will help the reader to visualize what amplitude differences the paper is talking about. The tuning curves are provided in column ii, but they are average over environments and I think it would be useful to clearly depict how the amplitudes change using some example cell responses.

We attempted to add some examples as requested. However, we struggled with the fact that the addition interrupted the flow of the text, and we were not sure the examples added critical information relevant to the conclusions of the manuscript – as these have to do with the norm of the population response.

Moreover, we are making **all the raw data** publicly available with the publication of the manuscript. The examples the reviewer requested can be easily generated with just one line of Matlab code after loading any of the datasets. Specifically, to plot the tuning curve of any cell in the population for all 3 environments, as requested, one can simply do:

```
>> plot(results.rs(:, :, n)')
```

Where n is the index of the desired neuron.

Thus, we decided against the proposed changes.